

# Comparison of approaches to interpolating climate observations in steep terrains with low-density gauging networks

Juan Ossa-Moreno[1], Greg Keir[1], Neil McIntyre[1], Michela Cameletti[2], and Diego Rivera[3]

[1]Centre for Water in the Minerals Industry, Sustainable Minerals Institute, The University of Queensland, Australia
[2]Department of Management, Economics and Quantitative Methods, Universita degli Studi di Bergamo, Bergamo, Italy
[3]School of Agricultural Engineering, Water Research Centre for Agriculture and Mining (WARCAM), Universidad de Concepcion, Concepcion, Chile

**Correspondence:** Juan Ossa-Moreno (j.ossamoreno@uq.edu.au)

**Abstract.** The accuracy of hydrological assessments in mountain regions is often hindered by the low density of gauges, coupled with complex spatial variations in climate. Increasingly, spatial data sets (i.e. satellite and gridded products) and new computational tools are used to address this problem, by assisting with the spatial interpolation of ground observations. This paper presents a comparison of approaches of different complexity to spatially interpolate precipitation and temperature time-series in the upper Aconcagua catchment in central Chile. A Generalised Linear Mixed Model whose parameters are estimated through approximate Bayesian inference is compared with three simpler alternatives: Inverse Distance Weighting, Lapse Rates and a method based on WorldClim data. The assessment is based on a leave-one-out cross validation, with the Root Mean Squared Error being the primary performance criterion for both climate variables, while Probability of Detection and False Alarm Ratio are also used for precipitation. Results show that for spatial interpolation of the expected values of temperature and precipitation, the WorldClim approach may be recommended as being the more accurate, easy to apply and relatively more robust to tested reductions in the number of estimation gauges, particularly for temperature. The Generalised Linear Mixed Model has comparable performance when all gauges were included, but is more sensitive to the reduction in the number of gauges used for estimation, which is a constraint in sparsely monitored catchments.

## 1 Introduction

Climate variables such as temperature and precipitation are key inputs for hydrological modelling and water resources management, and generally, spatial interpolation of point observations is desirable to support detailed analyses. Many interpolation approaches perform well for gentle terrains, however, their accuracy and precision decreases in mountain areas (Wu and Li, 2013; Frei, 2014; Buytaert et al., 2006; Falvey and Garreaud, 2007). As highlighted by Dorninger et al. (2008), challenges include observation errors, anisotropic climate patterns and sensitivity of results to density and location of observations. Strongly non-linear relations between temperature and altitude may be related to physiographic features (Stahl et al., 2006; Diodato, 2005), to cold-air trapped in enclosing hill ranges (Frei, 2014), and also to the presence of glaciers (Ragettli et al., 2014; Petersen and Pellicciotti, 2011). For precipitation, non-linearity can be related to physiographic features (Daly et al., 2008), to





the interaction between topography and rain-storms (Falvey and Garreaud, 2007; Garreaud, 2013; Viale and Garreaud, 2015; Diodato, 2005) and to summertime convective precipitation events (Viale and Garreaud, 2014).

These effects can be incorporated into spatial interpolation through deterministic approaches (Frei, 2014; Masson and Frei, 2014; Thornton et al., 1997, 2014; Hasenauer et al., 2003), inclusion of physiographic factors (Daly et al., 2002, 2008), geo-statistics (Wu and Li, 2013; Goovaerts, 2000), and other stochastic models (Aalto et al., 2013; Kenabatho et al., 2012). The quality of the outputs of these approaches often depends on the reliability and accuracy of climate gauges, which in many catchments are sparsely situated and/or of short record length. As a consequence, there is an increasing interest on alternative sources of data beyond point observations such as satellite and other gridded products (Dinku et al., 2014; Manz et al., 2016; Zambrano-Bigiarini et al., 2016; Dinku et al., 2010; Hobouchian et al., 2017; Demaria et al., 2013; Hijmans et al., 2005).

The Andes Cordillera in South America is an example of a steep terrain with sparse ground data and complex weather conditions. This mountain range is an important source of natural resources, including water for agriculture, mining and other industries. The stream-flows in the region are highly variable in both time and space (Pellicciotti et al., 2007; Mernild et al., 2017, 2016; Montecinos and Aceituno, 2003; Wolter and Timlin, 2011; Viale and Garreaud, 2014), therefore under such circumstances, quality of spatial climate data is a key issue when modelling water resources (Zambrano-Bigiarini et al., 2016; Mernild et al., 2017; Garreaud et al., 2017; Demaria et al., 2013).

This paper compares four approaches for interpolating temperature and precipitation in the upper section of the Aconcagua River (in the Chilen Andes). Despite being quite sparsely gauged compared to some mountain ranges globally (e.g. Swiss Alps), in the context of the central and southern Andes, this catchment has an unusually high number of gauges in high elevation points. The methodologies used include a Generalised Linear Mixed Model (GLMM - a spatio-temporal model) (Faraway, 2016), whose parameters were estimated using approximate Bayesian inference (Rue et al., 2009). This approach is relatively common in the statistics literature but is rarer in the hydrology realm.

Also applied are more commonly used (in hydrological and water resources modelling) deterministic approaches for inter-polation, including Lapse Rates (LR) (Ragettli et al., 2014)) and Inverse Distance Weighting (IDW). Finally, it was also used a method based on merging gauged data with WorldClim climate maps (Hijmans et al., 2005)).

The aim of the paper is to compare the performance of the approaches in a mountainous catchment with steep and complex topography. It specifically expects to test (1) the applicability of a GLMM whose parameters are estimated through approximate Bayesian Inference in a hydrological context, (2) compare the quality of outcomes of the four approaches and their sensitivity to the reduction of available gauges and (3) the added value of including alternative data sources. A full rationale for the selection of the GLMM and the approximate Bayesian Inference method is provided later in the paper.

The paper includes a description of the case study, the methods and the alternative sources of data. This is followed by a description of results, a discussion of the latter and the conclusions.





## 2  Case study and input data

### 2.1  Aconcagua River

The Aconcagua River is an important source of water in Central Chile (Pellicciotti et al., 2007). The source is located in the Andean mountains in the border between Chile and Argentina, and the river flows west towards the Pacific Ocean. The region is semiarid with a yearly average precipitation of around 350 mm concentrated during the austral winter (frontal rainstorms), when the South Pacific Anticyclone retreats from the region (Falvey and Garreaud, 2007). In addition, there is considerable inter-annual variability related to El Niño and La Niña phases (Montecinos and Aceituno, 2003). Topography fluctuates from coastal areas to peaks of elevations around 6000 m above sea level. The whole catchment has an area of around $7500 km^2$; however the upper section, which is the subject of this research, is only around a third of this and includes the Andean mountains and a portion of the central valley (see Figure 1).

Water resources management in some sub-catchments of the case study has received attention from researchers (Ragettli and Pellicciotti, 2012; Ragettli et al., 2014), who highlighted the importance of properly modelling snow accumulation and melting, which in turn requires accurate estimates of precipitation and temperature. This is particularly important for analysing potential impacts of changes in climate conditions (Pellicciotti et al., 2014; Vicuña et al., 2011) on economic activities.

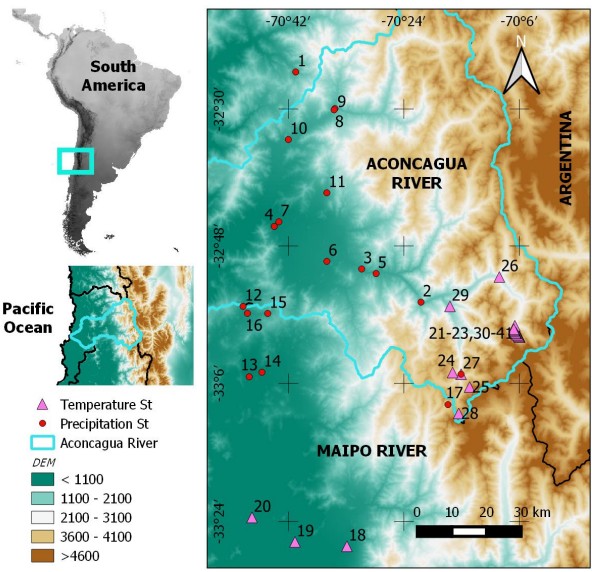

**Figure 1.** Temperature and precipitation gauges in the catchment with available data during the period of analysis. The numbers correspond to the numeration in the Appendix.



## 2.2   Precipitation and temperature gauges

Observations of daily average temperature and precipitation in the catchment were sourced from the Chilean General Water Directorate (DGA) and the Chilean Meteorological Directorate (DMC), through the Chilean Centre for Climate and Resilience Research (CR2) databases. Most of these gauges are located in lowlands, whereas the mountain areas are sparsely monitored

with the only available gauges from operational mine sites in the area. Amongst these gauges, there are two that record liquid and solid precipitation (Lagunitas and Los Bronces, see the Appendix for more details). The latter were transformed to snow water equivalents (SWE) before being analysed here.

This data was complemented with information from Universidad de Chile (Ohlanders et al., 2013) (available for some months only) and with measurements done by ETH-Zurich in the 2008-2009 summer season (Ragettli and Pellicciotti, 2012;

Pellicciotti et al., 2010). The latter was available during a very short period, but the measurements were done in an area different than the mine sites and nearby a major glacier, thus they provide valuable information to test the interpolation approaches. The location of the temperature and precipitation gauges is shown in Figure 1, while further details of the gauges (including the periods available and the percentage of missing values) are provided in the Appendix.

The period of analysis spans from September 2008 to August 2013 because this was the period with more data available,

as some gauges started or stopped recording measurements during these years (Figure 2 and Figure 3 show some of the data). Although not long enough to analyse long-term trends, the selected period allows testing the interpolation approaches over both dry and wet years (the average yearly precipitation of the gauges analysed during this period was 217 mm).

Quality control of climate data was done by analysing double mass plots and Pearson correlation values with patron gauges (e.g. long-term gauges previously used by academic and government sources (Jacquin and Soto-Sandoval, 2013; Ragettli et al.,

2014; Correa-Ibanez et al., 2017)). Beyond some issues with precipitation measurements in Hornitos and Saladillo (it was decided not to include these gauges in the precipitation Analysis), no further issues with data quality were noted.

## 2.3   Spatially distributed data sets

To complement the point observations, the Climate Hazards Group InfraRed Precipitation with Station data (CHIRPS) satellite product (Funk et al., 2015) was used. Including remotely sensed data to analyse climate variables is increasingly popular

amongst researchers, and several examples exist for precipitation in the Andes (Dinku et al., 2010; Zambrano-Bigiarini et al., 2016; Manz et al., 2016; Álvarez-Villa et al., 2011) and beyond (Nikolopoulos et al., 2013; Thiemig et al., 2012; Dinku et al., 2014). Based on these experiences in mountain regions, it could be said that satellite products tend to be good at detecting precipitation, but struggle to predict the magnitudes of the events, particularly for extreme conditions, and for daily and subdaily resolutions. This is usually a consequence of orographic effects and convective precipitation events.

Despite this, merging satellite data is one of the most promising options for spatial interpolation of precipitation, and catchment-specific studies are needed to develop this potential (Zambrano-Bigiarini et al., 2016). CHIRPS was chosen for this case study because its resolution (daily values with $0.05^o$ pixels) was ideal for the size of the case study, and because





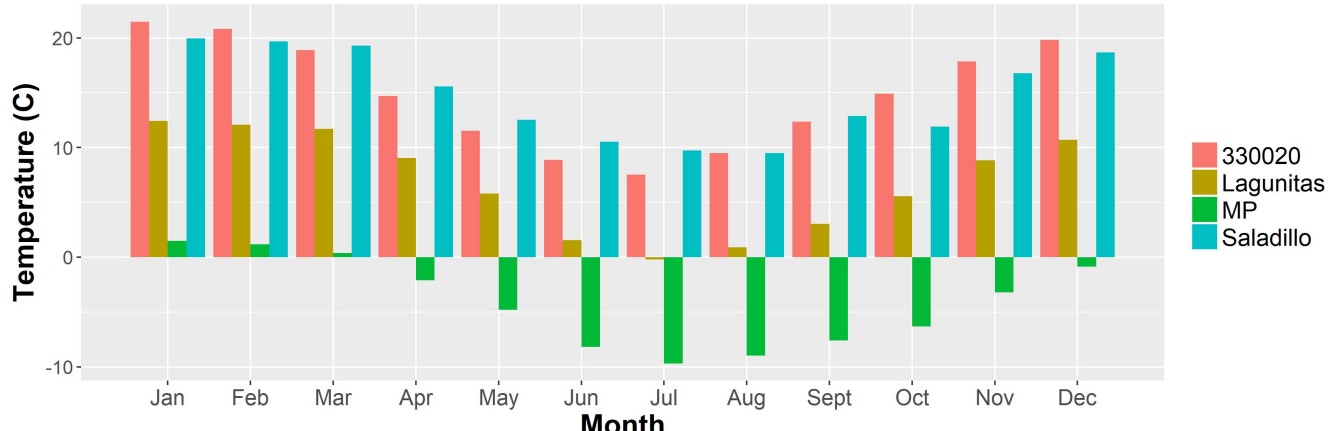

**Figure 2.** Monthly average temperature aggregated over the period of analysis for four of the gauges in the catchment: 330020 (527 masl), Saladillo (1580 masl), Lagunitas (2765.5 masl) and MP (4250 masl). The location of the gauges can be checked in Figure 1 and the Appendix.

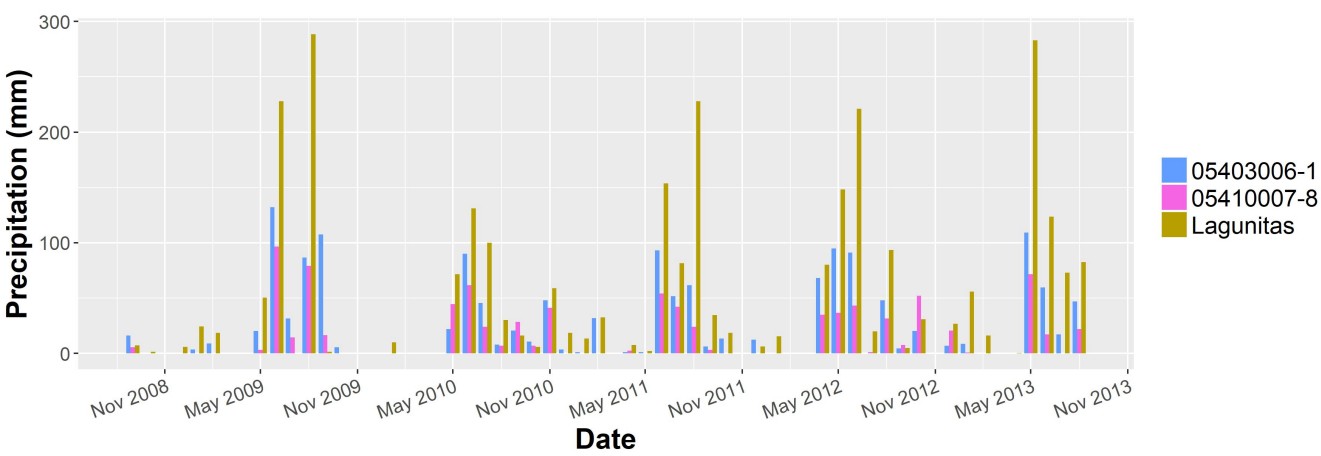

**Figure 3.** Monthly precipitation between 09/2008 and 08/2013 for three of the gauges in the catchment: 05410007-8 (820 masl), 05403006-1 (1290 masl) and Lagunitas (2765.5). The location of the gauges can be checked in Figure 1 and the Appendix.




recent studies showed good performance in Chile (Zambrano-Bigiarini et al., 2016). A sample image of CHIRPS is presented in Figure 4.

Gridded products such as WorldClim (WC) Version 1 (Hijmans et al., 2005) provide another potentially valuable source of data (see Figure 5). These climate surfaces provide a historical average for each one of the 12 calendar months (one map for

every month), with a $1km$ spatial resolution.

WC data originates from an statistical analysis of weather observations worldwide between 1950 and 2000, through an algorithm included in the ANUSPLIN interpolation package (Hutchinson, 2004), using latitude, longitude and elevation as independent variables in a regression. The developers of this data warn about potential inaccuracies of WC in mountainous areas (Hijmans et al., 2005), therefore, the WC data were never used independently but only to complement point-observations,

or as a benchmark for testing other interpolation approaches.

Although different in essence, both WC and CHIRPS can be used as a complement to point observations to construct daily or monthly interpolated fields. None of the selected gauged data were used as input in the construction of WC or CHIRPS [1], furthermore the 5-year period of analysis here does not overlap with the period used to develop WC.

The third spatial data set used was a Digital Elevation Model (DEM) based on the Shuttle Radar Topography Mission

(SRTM) (Jarvis et al., 2008), with a spatial resolution of $90m$. Finally, although not spatially distributed, a multivariate ENSO (El Niño-Southern Oscillation) index was also included to analyse the inter-annual variability of precipitation in the catchment (Wolter and Timlin, 2011).

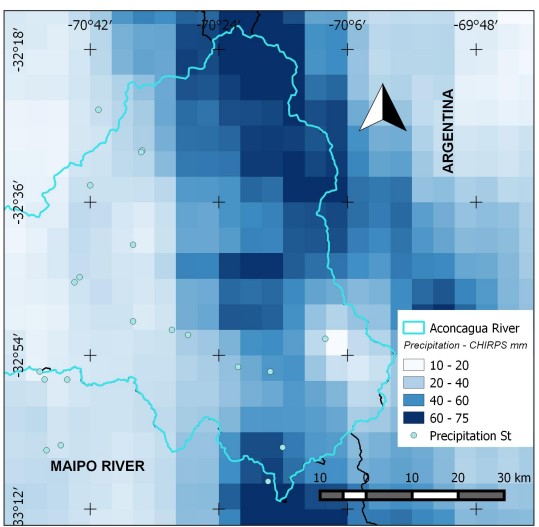

**Figure 4.** CHIRPS daily values aggregated for May 2009

---

[1]The name of the gauges used to calibrate CHIRPS can be checked here.





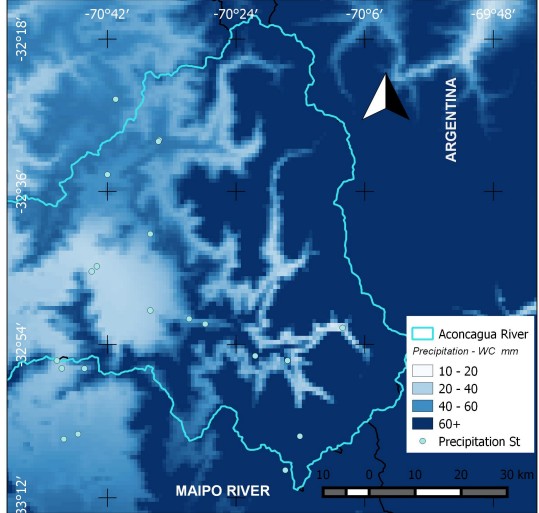

**Figure 5.** Worldclim precipitation - May (long-term average)

## 3 Analysis of Climate Data

The analysis and interpolation of climate variables in hydrology is done using a wide range of approaches, including simple methods such as Inverse Distance Weighting (Lu and Wong, 2008; Chen and Liu, 2012) and linear regressions (Ragettli et al., 2014; Ragettli and Pellicciotti, 2012; Meza et al., 2014; Masson and Frei, 2014). Other approaches like non-linear functions

and Generalised Linear Models (GLM) (Frei, 2014; Aalto et al., 2013) have also been used, sometimes including parameters that analyse the spatial correlation between observations (i.e. inter-site dependency) (Kenabatho et al., 2012; Kigobe et al., 2011; Chandler and Wheater, 2002). The Kriging family of methods, borrowed from the geostatistics literature, has also been widely used to analyse climate variables (Nerini et al., 2015; Álvarez-Villa et al., 2011; Benavides et al., 2007; Yao et al., 2013) (a more detailed review of examples can be seen in Li and Heap (2014); Bivand et al. (2013)). However, as in many other fields

beyond statistics, Generalised Linear Mixed Models (GLMM - also defined as Generalised Linear Mixed Effects Models) are less frequent.

GLMMs allow the analysis of non-normal observations as GLMs do, but the former are an extension of the latter due to their larger flexibility to analyse random effects (Faraway, 2016). GLMMs are frequently specified by means of a set of equations connected hierarchically; for this reason they are also known as multilevel or hierarchical models (Kéry and Royle, 2015; Rue

et al., 2009). GLMMs can be estimated using Bayesian or likelihood-based approaches; the former being adopted in this paper. In both cases, they avoid using the *method of moments* to define empirical/experimental variograms (Minasny and McBratney, 2005), and the subsequent adjustment of a theoretical variogram through a curve-fitting exercise (Ecker and Gelfand, 1997; Müller, 1999), as sometimes done for Kriging applications in hydrology (Goovaerts, 2000; Nerini et al., 2015).





The main drawback of analysing GLMMs with the Bayesian approach is the computational requirements of the classical simulation-based methods such as Markov Chain Monte Carlo (MCMC) (Cameletti et al., 2011), and this is perhaps why they are less attractive compared to simpler alternatives in fields like hydrology. However, the relatively recent Integrated Nested Laplace Approximation together with the Stochastic Partial Differential Equation approach (INLA-SPDE) (Rue et al., 2009; Lindgren et al., 2011; Cameletti et al., 2013), represents a computationally efficient way to do approximate Bayesian inference on GLMMs and other models belonging to the class of latent Gaussian models (Rue et al., 2009).

### 3.1 The GLMM in the Aconcagua case study

The climate variables in the case study (temperature and precipitation) are assumed to be realisations (e.g. observations) of a spatio-temporal process (random field) of the form:

$$Y(s,t) \equiv \{y(s,t), (s,t) \in \mathbb{D} \subseteq \mathbb{R}^2 \times \mathbb{R}\} \tag{1}$$

where $s$ and $t$ denote the spatial location and time. This process has a mean $\mu$ and covariance function $Cov(y(s,t), y(s',t')) = \sigma^2 C((s,t),(s',t'))$ (Blangiardo et al., 2013; Cameletti et al., 2013). Assuming that climate observations, $\mathbf{y} = \{y(s_i,t), i = 1,...,N, t = 1,...,T\}$, follow an exponential family probability distribution function (PDF), $\mu_i$ can be connected to a structured additive predictor $\eta_i$ through a link function $g(\ )$ as shown below (Rue et al., 2009):

$$g(\mu(s_i,t)) = \eta(s_i,t) = \alpha + \sum_{j=1}^{n_f} f^{(j)}(u_{j(s_i,t)}) + \sum_{k=1}^{n_\beta} \beta_k z_{k(s_i,t)} + \epsilon(s_i,t) \tag{2}$$

$$\mathbf{x} = (\alpha, \{f^{(j)}(.)\}, \{\beta_k\}, \{\eta(s_i,t)\}) \tag{3}$$

$$\epsilon(s_i,t) \sim N(0, \sigma_\epsilon^2) \tag{4}$$

where $\mathbf{x}$ is the vector including the Gaussian latent processes (i.e. the parameters describing the random field), $\epsilon(s_i,t)$ is the random error component, the $f^{(j)}(u_{j(s_i,t)})$ are functions of covariates $u$ and the $\beta s$ are the multipliers of covariates $z$.

For temperature, the model in this project was defined based on the one described in Cameletti et al. (2013) and Cameletti et al. (2011) for particulate matter, with daily time-steps. This selection was done taking into account that both variables are affected by their values in previous time-steps, but also because both of them have a spatial correlation. The model is described as follows:

$$y(s_i,t) = z(s_i,t)\beta + \xi(s_i,t) + \epsilon(s_i,t) \tag{5}$$


$$\xi(s_i,t) = a\xi(s_i, t-1) + \omega(s_i,t) \tag{6}$$

where $y(s_i,t)$ represents a realisation of the gaussian field (GF) $Y(.,.)$ for site $s_i$ and time $t$, $z(s_i,t) = (z_1(s_i,t),...,z_p(s_i,t))$ are the covariates (fixed effects), $\beta$s are the coefficients of the covariates, $\epsilon$ is the measurement/observation error component, both serially and spatially uncorrelated ($\epsilon(s_i,t) \sim N(0,\sigma_\epsilon^2)$), and $\xi$ represents the random component in the model. The latter is defined as a first-order autoregressive (AR) component with spatially correlated innovations $\omega(s_i,t)$. The covariates included latitude, longitude, elevation and WC. Data from WC maps were included in the model as covariates, after extracting the values of the pixels containing the gauges.

The spatio-temporal model for precipitation was defined based on previous experiences of applications of INLA-SPDE on GLMMs for this variable. This involved dividing the analysis into occurrence (Eq. 7) and magnitude (Eq. 8) components, based on Eq. 8.5 and Eq. 8.6 in Blangiardo and Cameletti (2015). However, it was decided to use monthly time-steps as preliminary results of daily runs were far from satisfactory. In addition, CHIRPS and the ENSO index were included as covariates to complement the ones used for temperature.

$$O(s_i,t) \sim Binomial(\pi(s_i,t),1) \tag{7}$$

$$y^P(s_i,t) \sim Gamma(a(s_i,t),b(s_i,t))$$
$$E(y^P(s_i,t)) = \mu^P(s_i,t) = a(s_i,t)/b(s_i,t) \tag{8}$$
$$Var(y^P(s_i,t)) = a(s_i,t)/b(s_i,t)^2$$

Dummy variables for each calendar month were included as additional covariates, in order to better represent the strong seasonality of precipitation in the case study (Falvey and Garreaud, 2007; Montecinos and Aceituno, 2003). In this way, the random process $\Phi(s_i,t)$ is spatially correlated but independent of other time-steps. The model is described as follows:

$$logit(\pi(s_i,t)) = z^P(s_i,t)\beta^P + \Phi(s_i,t) + \epsilon^P(s_i,t) \tag{9}$$

$$log(\mu^P(s_i,t)) = z^P(s_i,t)\beta^P + \epsilon^P(s_i,t) + \beta^{P'}\Phi(s_i,t) \tag{10}$$

The link functions connecting the mean of the GF and the predictors are not unitary, as for temperature, but $logit$ (occurrence) and $log$ (magnitude). Both Eq. 9 and Eq. 10 share the same $\beta^P$s, but the latter has an extra parameter ($\beta^{P'}$) connecting the random field in both equations.





It is acknowledged that many more models could be tested with these climate variables (e.g. as done in Cameletti et al. (2011) for PM10), and this represents a subject for future research. However, this project is focused on the comparison of the performance of GLMMs (whose parameters are estimated with INLA-SPDE) with simpler methods often used in hydrology, and on the inclusion of alternative data sets. Therefore, it was desired to work with GLMMs already available in the literature

(or close adaptations), which have been previously analysed with the INLA-SPDE approach.

## 3.2 Alternative approaches

The GLMM was compared to simpler deterministic approaches: IDW, LR (Pellicciotti et al., 2014; Ragettli et al., 2014), and a simple method developed in this project based on WC maps, which from now on is defined as WC Adjusting (WCA). It is assumed that the reader is familiar with IDW and LR. Briefly, the former estimates variables at unsampled locations $y(s_j, t)$ as

a function of the inverse of the distance $d(s_j, s_i)$ between $s_j$ and all sampled locations $s_i$ following

$$y(s_j, t) = \frac{\sum_{i=1}^{n} y(s_i, t) \frac{1}{d(s_j, s_i)}}{\sum_{i=1}^{n} \frac{1}{d(s_j, s_i)}} \qquad (11)$$

where $y(s_i, t)$ are the values at the $n$ sampled locations. This method does not consider elevation effects. LR, on the other hand, uses linear and logarithmic regressions to model the relation between temperature or precipitation and elevation. The regressions could be extended to include all the covariates of the GLMM, however, the objective here was to apply the methods

as they are commonly used to define inputs of hydrological and water resources models (Ragettli et al., 2014; Vicuña et al., 2011; Meza et al., 2014).

The WCA method attempts to couple the benefits of the spatial variability of WC maps and those of the temporal resolution of the observations. This method is described as follows:

1. For each time-step, each climate observation $y(s_i, t)$ was compared with the corresponding value in the WC maps

$WC(s_i, t)$ (i.e. the value of the pixel where the observation is located), to define a residual $R$. Assuming that in the centre left pixel of map **A** in Figure 6 there is a gauge, and that $WC(s_i, t) = 5.1(^oC)$ and **B** $y(s_i, t) = 7.1(^oC)$, the residual can be defined as $R = 2.0(^oC)$.

2. This residual $R$ was added to the WC map (map $A$ in Figure 6) to define an adjusted map (map $C$ in Figure 6). Steps 1 and 2 were repeated for all gauges available, in order to generate one adjusted map for each gauge at each time-step.

3. All adjusted maps for the same time-step were merged to define the values at all pixels (see Figure 7 for an example of the top left pixel). This was done using inverse distance weighting, between the location of the gauges and the pixel being interpolated (the example in Figure 7 uses 2 gauges). The same process is done for temperature and precipitation.

A summary of all interpolation approaches is provided in Table 1. For precipitation, due to the spatial smoothing that is inherent to all approaches, it is common to have very low values of precipitation where none is observed. Therefore, a threshold

of 1 mm/month was set, below which all values were deemed to be 0.

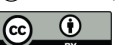


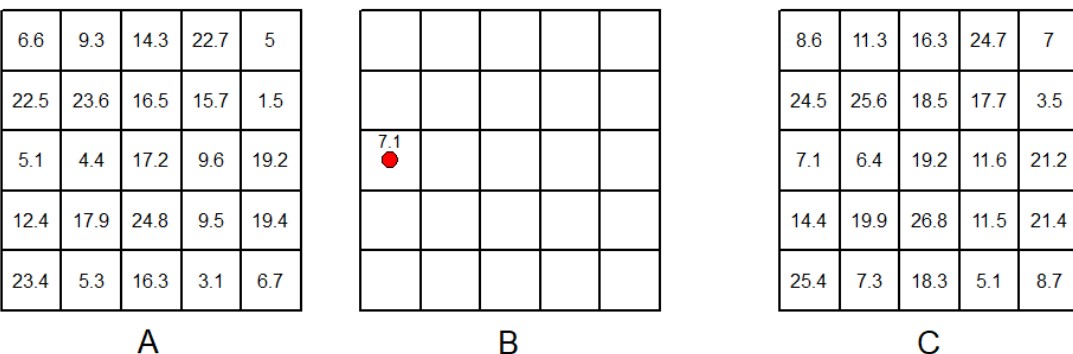

**Figure 6.** Step 1 Compares a WC value (A) with the observed value (B) to define the residual, which is added to the original WC map (A) to define an adjusted WC map (C) in step 2. This process is repeated for every gauge available at each time-step.

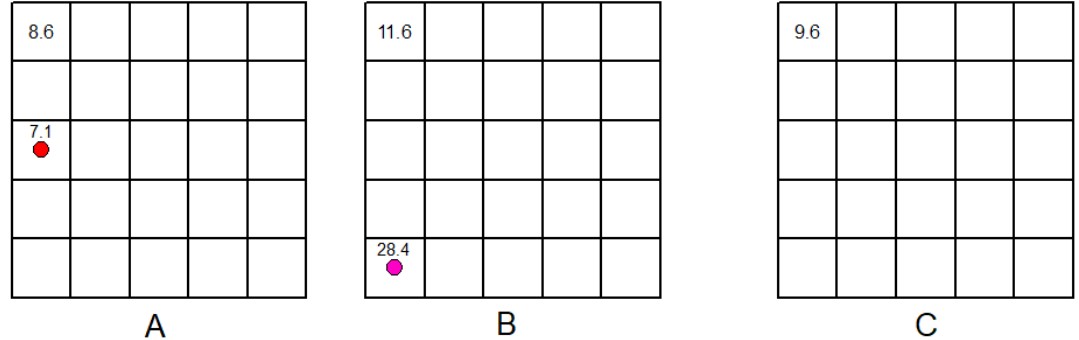

**Figure 7.** (A) shows two pixels of map C in Figure 6. (B) Shows the observed value of a second gauge in the bottom left corner (28.4 $^{\circ}C$) and the adjusted value in the upper left corner (11.6 $^{\circ}C$), after applying Steps 1 and 2. Taking into account that distance from gauge A (red dot) to the upper left pixel is half of that from gauge B (pink dot), Step 3 uses inverse weighting to define the merged value (9.6 $^{\circ}C$). This is repeated for all pixels first, and then for all time-steps.

The approximate Bayesian inference approach (INLA-SPDE) used to estimate the parameters of the GLMM was run using the INLA package for R (Rue et al., 2013), and this required using the Euramoo and Flashlite High Performance Computers (HPC) system from the Queensland Cyber Infrastructure Foundation (QCIF). All other interpolation approaches were run on a computer with 16 Gb of memory, an i7 processor and 4 cores.



**Table 1.** Summary of approaches to interpolate climate variables.

| Approach | Description | Error Model | Temporal Correlation |
|---|---|---|---|
| IDW | Interpolation based on the inverse of the distance | No random component | The method is run for every time-step independently |
| LR | Interpolation based on regressions using elevation as independent variable | No random component | The method is run for every time-step independently |
| WCA | Interpolation based on residuals of observations and values in WC maps | No random component | The method is run for every time-step independently |
| GLMM | Spatio-temporal model whose parameters are estimated through approximate Bayesian inference | First order autoregressive process with spatially correlated innovations for temperature and spatially correlated innovations for precipitation | AR1 for temperature and monthly dummy variables for precipitation |

## 3.3 Comparison of interpolation approaches

A leave-one-out cross-validation (LOOCV) (Manz et al., 2016) method was used to assess the performance of the approaches for both temperature and precipitation. Then, the sensitivity of performance to the number of gauges used for estimation was tested.

For temperature there were a total of 24 gauges available, thus, the LOOCV analysed the 24 combinations of 23 gauges, leaving one at a time for validation. For the sensitivity analysis it was only used the 9 gauges with relatively large observation periods (i.e. 15 out of the 24 gauges had observations for the 2008-2009 summer season only - see Figure 1 around $-70^{o}06'$ longitude and $-33^{o}00'$ latitude). Including the 24 gauges in the sensitivity analysis would have been a problem when using a reduced number of gauges for estimation, as several combinations would have had no data for most of the period of analysis.

In this way, the performance of all approaches was tested by using all combinations of 8 gauges to estimate results, and using the remaining gauges of each combination (plus the 15 with few data) for validation purposes. The overall metric was the average result of the validation groups. This was then repeated for all combinations of 5 and 2 gauges.

     For precipitation, there were 18 gauges available during most of the period of analysis, however only two of them were located in the mountains. The same procedure was followed for this variable for the LOOCV (18 combinations of 17 gauges)

and for the sensitivity analysis, but this time the latter was done with 14 and 4 gauges.

     For all tests, the average Root Mean Squared Error (RMSE) of the validation group was used to assess the performance of temperature and precipitation predictions, following similar comparisons (Cameletti et al., 2013; Manz et al., 2016; Nerini et al., 2015). For the GLMM, this involved analysing the expected values of each variable for each time-step. This was complemented with an analysis of the distribution of the residuals of the validation groups of the LOOCV. Furthermore, two categorical





statistics, the False Alarm Ratio (FAR) and the Probability of Detection (POD), were used to assess to what extent the model is able to predict precipitation occurrence (see Table 2) (Castro et al., 2015; Tobin and Bennett, 2012).

**Table 2.** Categorical statistics used to assess the capacity of the interpolation approaches to predict the occurrence of precipitation.

| *Precipitation* | Observed | Not Observed |
|---|---|---|
| Predicted | A | B |
| Not Predicted | C | D |
| **POD** | $\frac{A}{A+C}$ | |
| **FAR** | $\frac{B}{A+B}$ | |

## 4 Results

Before starting the interpolation of variables, their correlation with the covariates was assessed. As expected a priori, monthly temperature values were inversely correlated to elevation (Pearson Correlation Coefficient $\rho = -0.81$ - see Figure 9). There was also a strong correlation between WC values and monthly temperatures ($\rho = 0.98$, see Figure 8). Daily temperature values showed considerable correlation with elevation ($\rho = -0.77$) and WC ($\rho = 0.93$) as well. ENSO showed a low correlation with temperature ($\rho = 0.04$), thus it was decided not to include this covariate in this model.

The correlation between CHIRPS and daily precipitation observations was weak (see Figure 10), but considerably improved when both were aggregated to monthly values ($\rho = 0.81$ - see Figure 11). The $\rho$ for monthly precipitation and WC values was lower ($\rho = 0.62$, see Figure 12), while monthly correlation with elevation was above 0.6 for most months (see Figure 13). ENSO showed a weak correlation with precipitation ($\rho = 0.12$), however, a monthly analysis showed that for several months the correlation was close to $\rho = 0.5$, therefore, it was decided to keep ENSO as a covariate for the precipitation model. These correlations may be stronger in longer-term analyses that cover several Niño-Niña cycles, which last around 2-5 years each (Wolter and Timlin, 2011; Garreaud et al., 2017; Montecinos and Aceituno, 2003).

### 4.1 Temperature results

Table 3 shows the results of all interpolation approaches in terms of the average RMSE of the validation gauges in the LOOCV (23 gauges). It was found that the GLMM and WCA have the best performance, while LR and particularly IDW have larger RMSE values.

In addition, Table 3 also shows the results of the validation groups of all combinations in the sensitivity analysis. As expected a priori, it can be seen that errors increase when the number of estimation gauges decreases. However, values for WCA increase the least, and its loss of performance is relatively small even when only two estimation gauges are used. On the other hand, the performances of all other approaches, including the GLMM, show a sharp decline, to the point that some of their RMSE values are comparable with the range of observed temperatures (see Figure 2).




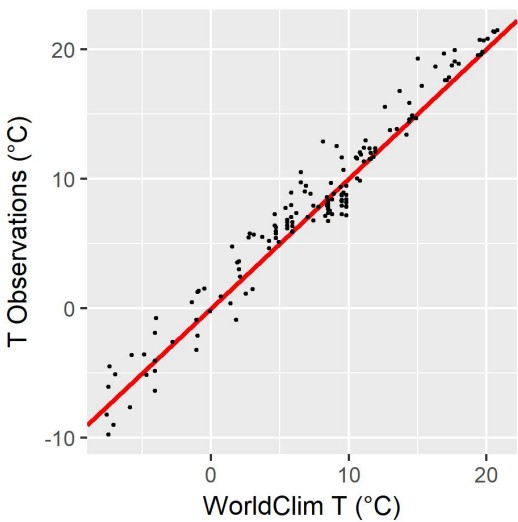

**Figure 8.** WC values vs. monthly aggregated (averaged) temperature values. The red line correspond to the 1:1 line.

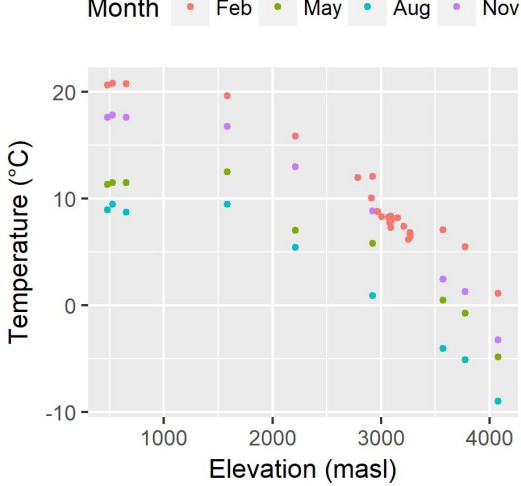

**Figure 9.** Elevation of gauges vs average temperature in four months.



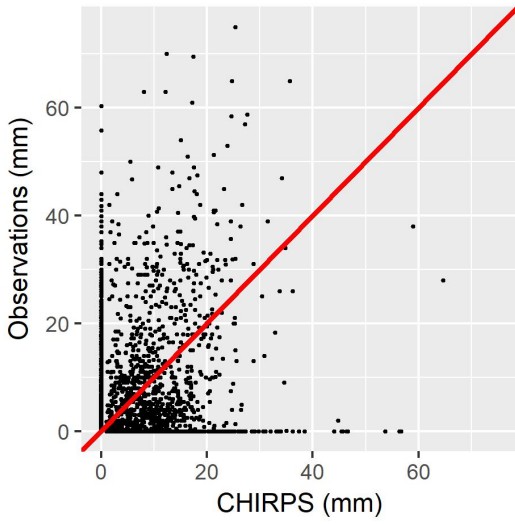

**Figure 10.** CHIRPS vs precipitation. Daily values for all stations used. The red line correspond to the 1:1 line.

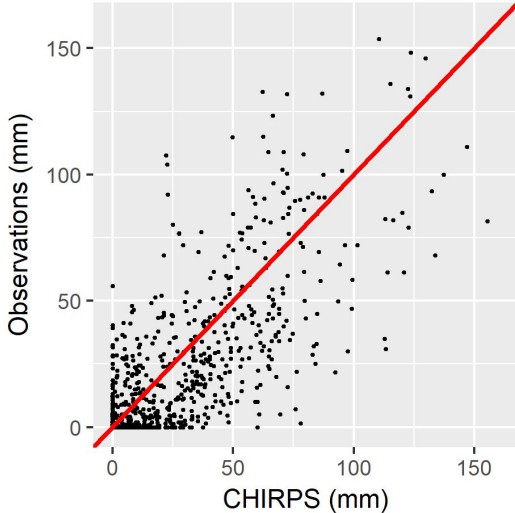

**Figure 11.** Monthly aggregated (sum) CHIRPS vs monthly aggregated precipitation values. The red line correspond to the 1:1 line.





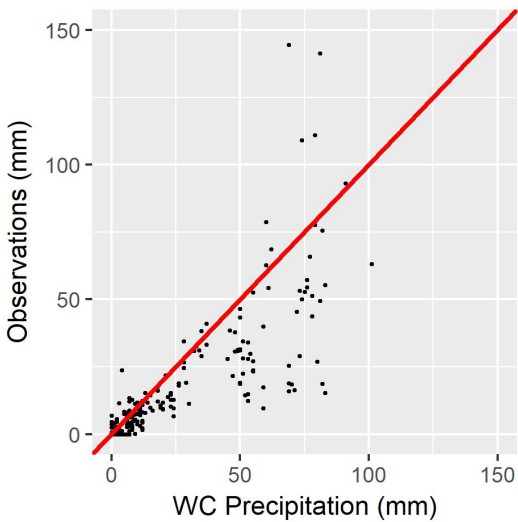

**Figure 12.** WC values vs. monthly aggregated (sum) precipitation values. The red line correspond to the 1:1 line.

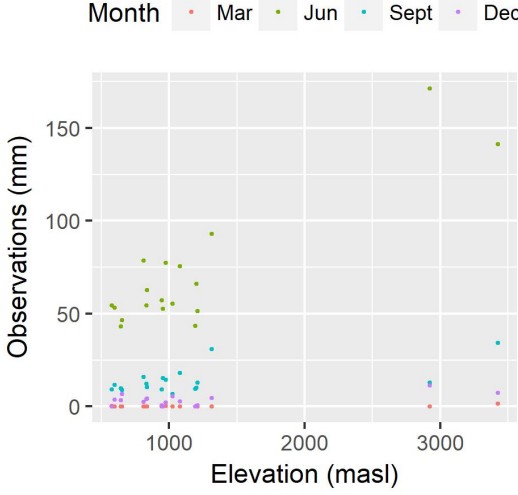

**Figure 13.** Elevation of gauges vs average precipitation in four months.





**Table 3.** Average RMSE in the leave-one-out cross validation for each interpolation approach.

| Approach | Number of estimation gauges | RMSE ($^oC$) |
|---|---|---|
| GLMM | 23 (LOOCV) | 1.2 |
| | 8 | 3.89 |
| | 5 | 3.99 |
| | 2 | 14.44 |
| WCA | 23 (LOOCV) | 1.22 |
| | 8 | 1.77 |
| | 5 | 1.98 |
| | 2 | 2.54 |
| | 0 (Raw WC Maps) | 3.36 |
| LR | 23 (LOOCV) | 1.53 |
| | 8 | 2.12 |
| | 5 | 4.14 |
| | 2 | 7.78 |
| IDW | 23 (LOOCV) | 2.72 |
| | 8 | 4.42 |
| | 5 | 6.15 |
| | 2 | 9.34 |

Figures 14, 15 and 16 show the daily temperature averaged over the 5-year period of analysis for three validation gauges (18, 27 and 28 in Figure 1 - similar results were found for the rest of them). It can be seen that the GLMM and WCA manage to reproduce the values from most of them relatively well, except for the MP gauge (the one at the highest elevation - 4250 masl) where larger differences can be seen. LR and particularly IDW tend to underestimate values in all gauges, except for MP, which they overestimate.

It is worth mentioning that in Figure 15, the overestimation of temperature for the LR method around March, was because during March 2009 all other gauges in the mine site stopped measuring, thus the predictions for Lagunitas were done with the lower elevation data only. This generated large errors for this gauge and this method, highlighting the issues with LR when few gauges are available for estimation purposes. This will be further discussed later in this section.

Figure 17 shows histograms of the validation residuals. It can be seen that the GLMM, WCA and LR have residuals that are more or less evenly distributed around zero, although those of the GLMM are more peaked. The distribution of IDW residuals is strongly multi-modal indicating consistent over or under-estimation at particular gauges. Furthermore, Figure 18 shows the relationship between RMSE values and elevation.




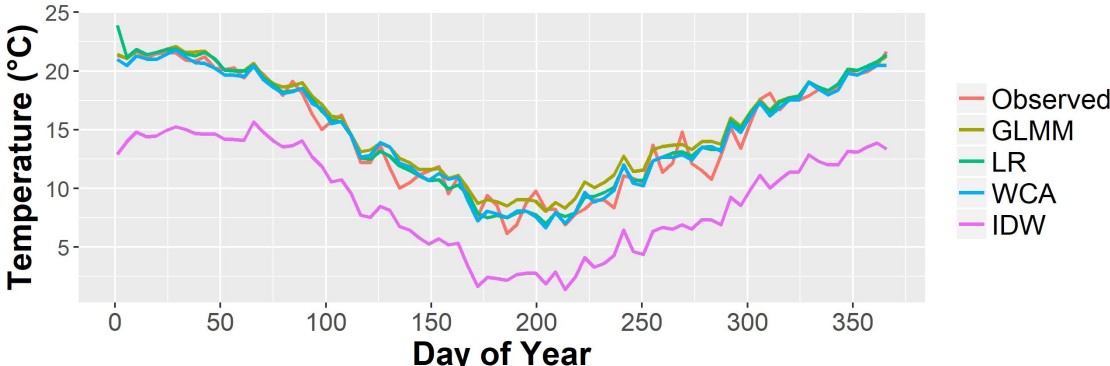

**Figure 14.** Daily temperature averaged over the 5 years of analysis for gauge 330019 (All curves were smoothed using the LOESS method ((Jacoby, 2000)) with $\alpha = 0.045$ to facilitate visualisation).

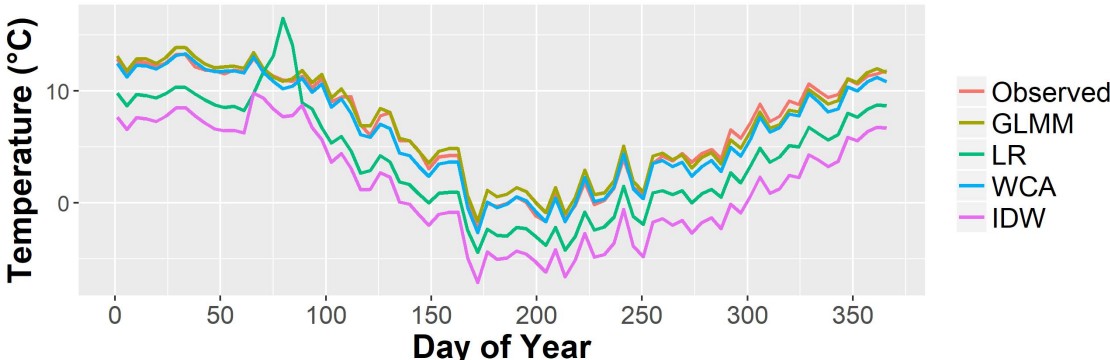

**Figure 15.** Daily temperature averaged over the 5 years of analysis for Lagunitas (All curves were smoothed using the LOESS method ((Jacoby, 2000)) with $\alpha = 0.045$ to facilitate visualisation).

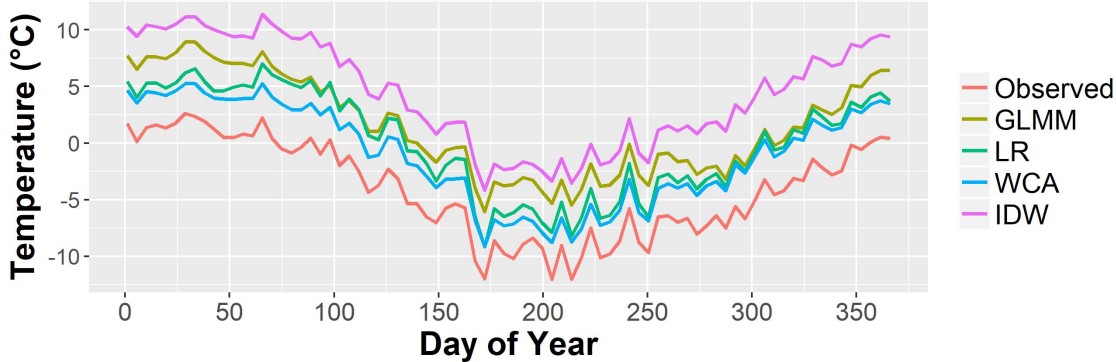

**Figure 16.** Daily temperature averaged over the 5 years of analysis for MP (All curves were smoothed using the LOESS method ((Jacoby, 2000)) with $\alpha = 0.045$ to facilitate visualisation).

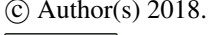



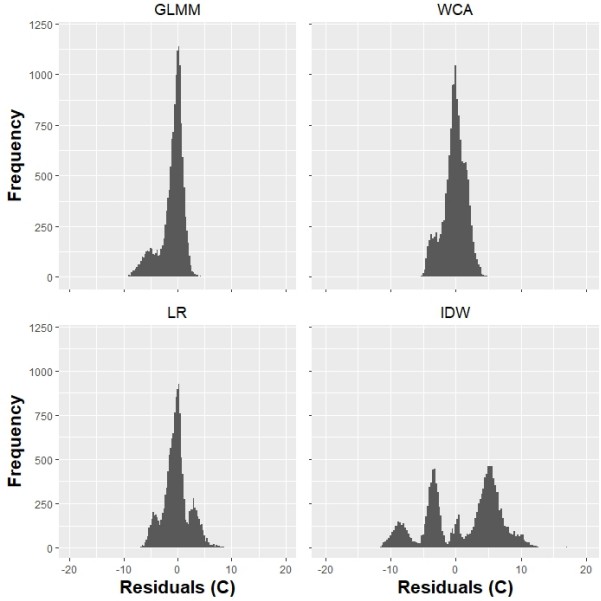

**Figure 17.** Residuals of the temperature LOOCV for each interpolation approach.

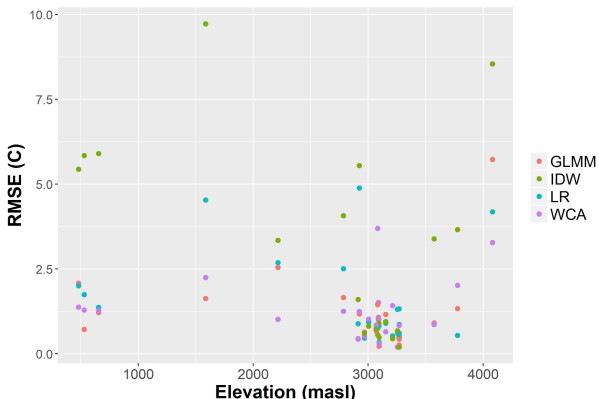

**Figure 18.** Elevation vs Temperature RMSE for all gauges in the validation groups of the LOOCV.

## 4.2 Precipitation results

Table 4 shows that the performances of all interpolation approaches are similar in the LOOCV, in terms of RMSE, although WCA and IDW have slightly smaller RMSE values. All probability of detection (POD) indices are above 90%, although WCA and IDW have values closer to 100%. Differences in false alarm ratios (FAR) are larger, as the GLMM has a ratio of only 7.1%, which is almost half of the one for LR and less than a third of that of IDW and WCA.





Table 4 also shows how sensitive are the interpolation approaches to the reduction in the number of estimation gauges. It can be seen that the GLMM was quite sensitive to these changes, and its performance decreased sharply with 14 and particularly with 4 gauges in terms of the RMSE. Its POD and FAR remained similar to the values in the LOOCV. The other 3 approaches behaved similarly with 14 gauges, although LR had lower POD and FAR. With 4 gauges WCA shows the smallest increase in
RMSE, although its FAR has the largest increase. On the other hand, LR has a larger RMSE but a low FAR again.

When these values are compared with raw CHIRPS and WC values, it can be seen that the performance of both is not competitive when there are 17 or 14 gauges available. However, the quality of CHIRPS predictions is similar to results with only 4 gauges in terms of RMSE, POD and FAR. This suggests that if there were four or less gauges available in the catchment, using CHIRPS would be a useful alternative.

**Table 4.** Results of the approaches tested to interpolate precipitation.

| Approach | Number of estimation gauges | RMSE (mm) | POD (%) | FAR (%) |
|---|---|---|---|---|
| GLMM | 17 (LOOCV) | 14.2 | 92.3 | 7.1 |
| | 14 | 32.1 | 91.8 | 7.07 |
| | 4 | 135.8 | 87.8 | 10.6 |
| WCA | 17 (LOOCV) | 13.4 | 97.3 | 24 |
| | 14 | 17.4 | 97.5 | 25.8 |
| | 4 | 23.5 | 95.3 | 27.9 |
| | 0 (Raw WC Maps) | 34.1 | 98.6 | 40.5 |
| IDW | 17 (LOOCV) | 13.5 | 98 | 22.7 |
| | 14 | 17.8 | 97.2 | 22.1 |
| | 4 | 25.4 | 94 | 19.1 |
| LR | 17 (LOOCV) | 15.5 | 93.7 | 12.9 |
| | 14 | 18.9 | 90.6 | 15.7 |
| | 4 | 26.4 | 84.4 | 11.7 |
| CHIRPS | 0 (Raw CHIRPS data) | 26.2 | 88.5 | 28.6 |

Figures 19, 20 and 21 show the observed and simulated monthly precipitation values for three representative gauges. Figure 19 shows the performance of $05200007 - 6$, which is quite similar to that of all gauges in the lowlands. It can be seen that most approaches reproduced observed values relatively well (compared to the gauges in the mountains). Furthermore, it can also be seen that IDW and WCA predicted small amounts of precipitation in several months during the dry season when no precipitation was observed, which causes a larger FAR for both of them (see Table 4).

Figure 20 shows the performance of all approaches for Lagunitas, which is in the mountains at $2765\ masl$. In this plot it can be seen that observed precipitation is larger than in the lowlands, and that all approaches fail to reproduce observations with the level of accuracy shown in Figure 19. Figure 21 illustrates results for Los Bronces gauge, the highest of the precipitation



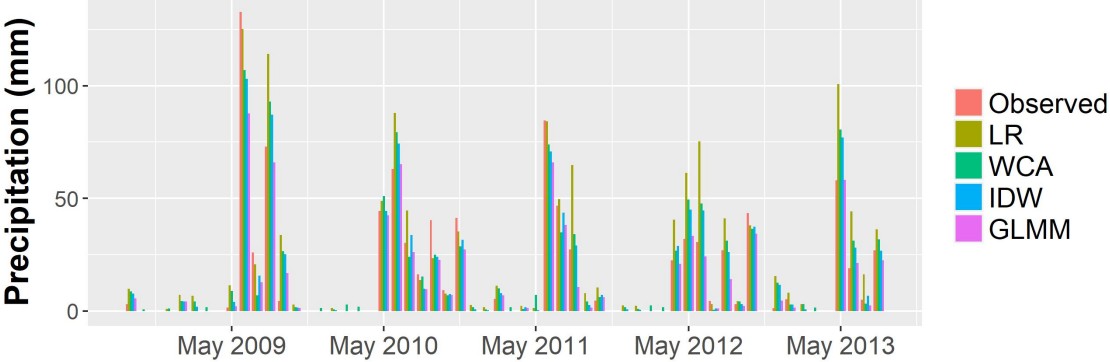

**Figure 19.** Validation monthly precipitation estimates for gauge 05200007-6.

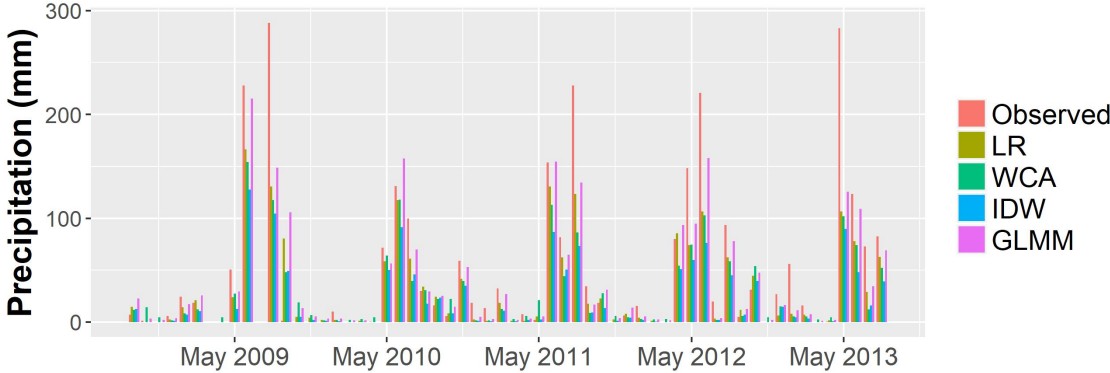

**Figure 20.** Validation monthly precipitation estimates for Lagunitas.

gauges (3420 $masl$). Once more, larger errors can be seen compared to the gauges in the lowlands, particularly for the GLMM, which nevertheless had the best results for Lagunitas.

This behaviour can be better appreciated after plotting the elevation of the gauges versus their average RMSEs (see Figure 22). While RMSE values below 1500 $masl$ are rarely above 20 mm, all the RMSE values of the two gauges above 1500 $masl$ are above this threshold, some of them are beyond 40 mm and two points are above 60 mm. This suggests that the performance of all approaches is likely to be determined by inaccuracies at high elevation gauges, where frontal systems interact with the topography to create very wet conditions during the wet season.

Regarding the residuals of all approaches (see Figure 23), as for temperature, the residuals using GLMM are more peaked around zero. Nevertheless, its greater number of very large residuals gives the GLMM a higher RMSE than WCA or IDW.





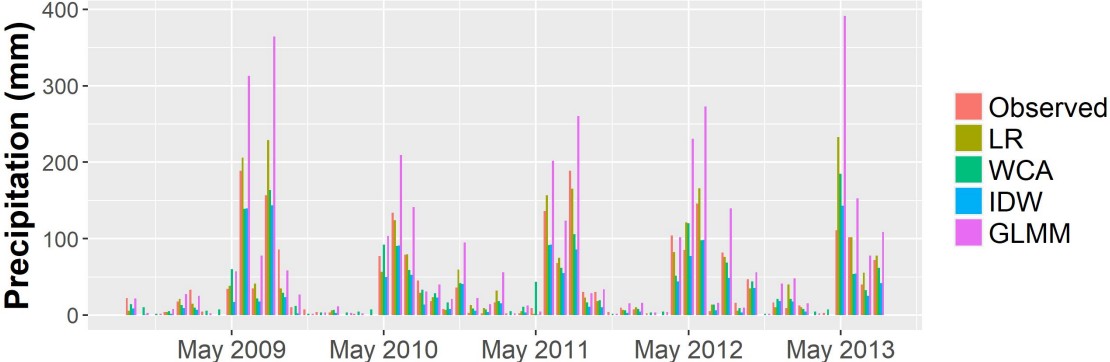

**Figure 21.** Validation monthly precipitation estimates for Los Bronces.

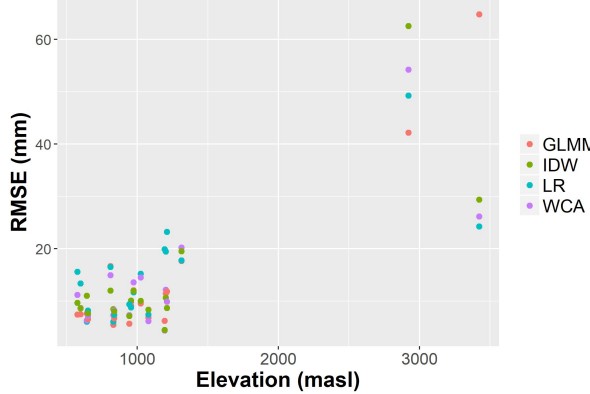

**Figure 22.** Elevation vs Precipitation RMSE for all gauges in the validation groups of the LOOCV.

## 5   Discussion

The LOOCV analysis of air temperature in Section 4.1 showed that for this case study, the GLMM and the WCA have the best performance (i.e. smallest RMSE values - see Table 3), although the magnitudes of LR results are also comparable with those obtained from similar analyses in USA and Canada (Stahl et al., 2006; Wu and Li, 2013). However, compared to the GLMM,

5  WCA has less computational requirements thus is easier to implement (i.e. WCA was run on a desktop computer as described in Section 3.2, while the GLMM was run on 20 HPC cores in parallel).

On the other hand, IDW had the largest errors and this, together with the skewed and multi-modal nature of its residuals, showed the limitations of this approach to analyse this variable. Figure 16 and 18 suggest that IDW residuals can sometimes be related to the high elevation (e.g. MP) or isolation (e.g. Saladillo) of gauges. The observations from the 2008-2009 summer

10  season have the best RMSE values for IDW, and this is likely to be due to the proximity and quantity of gauges in this period.

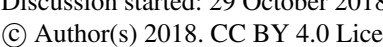
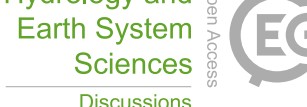


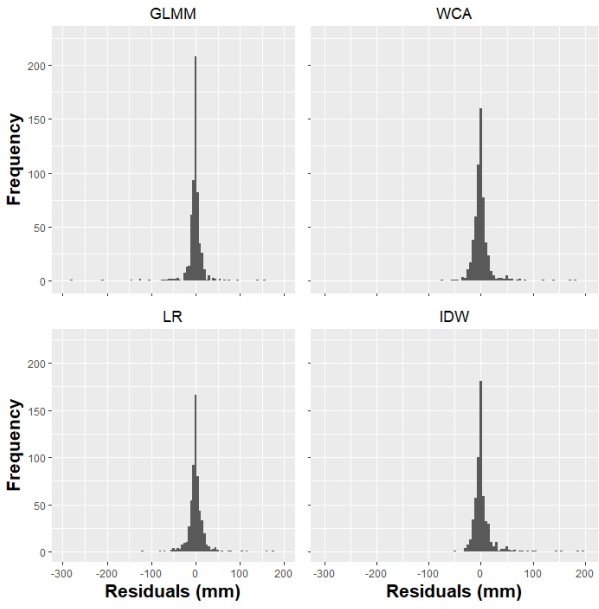

**Figure 23.** Residuals of the precipitation LOOCV for each interpolation approach.

The latter, however, could be a problem for the estimation of values of other gauges as IDW does the interpolation based on distances between gauges only, but does not necessarily analyse the effects of data clustering.

In terms of the influence of the elevation of gauges on results, WCA and LR showed similar performance across all elevations, but the GLMM had an outstanding error in the highest gauge (MP). This may suggest that compared to WCA and LR, this approach is more sensitive to the extrapolation of results beyond the altitude ranges of the estimation gauges.

Furthermore, it was found that the quality of results of the GLMM are sensitive to the number and location of gauges measuring temperature. As shown in Table 3, the RMSE for this approach rose considerably when only 8, 5 and 2 gauges were used to estimate its parameters. The performance of IDW and LR also decreased (RMSE of $9.34^oC$ and $7.78^oC$ respectively, with only two gauges), to the extent that using estimates of long-term monthly average temperatures provided by the WC maps would be a better alternative (RMSE of $3.36^oC$).

On the other hand, WCA was quite resilient to the reduction of estimation gauges, and its errors were lower, even with two gauges only (RMSE of $2.54^oC$). This may be because the raw WC maps have internalised the average effect of elevation, longitude and latitude on this climate variable through the long-term analysis (a worldwide generalisation), which can then be adapted to local conditions by including a small number of gauges. This suggest that WCA is an accurate and easy to use alternative to model air temperature in the case study.

Regarding precipitation, the LOOCV showed that all approaches have a similar performance in terms of RMSE ($13 - 15mm$). However, the GLMM stands out due to its lower FAR ($7.1\%$), which may be a positive outcome of separating the analysis of precipitation into occurrence and magnitude. This could also be related to the fact that the GLMM analyses the correlation





between measurements, thus limits the possibility of one or few gauges with non-zero precipitation overly influencing the precipitation estimate at all points (e.g. smoothing).

As opposed to this, other alternatives, particularly WCA and IDW, tend to predict precipitation when at least one (IDW) or even when no gauges (WCA - due to the inclusion of long term averages) record non-zero values. This is evidenced with the

prediction of dry-season precipitation events that were never observed. Preliminary results obtained using a different threshold (0.3 mm) for the detection of precipitation were similar, thus, the preference for GLMM in terms of FAR and POD performance seems not to be sensitive to the selection of this threshold.

When the interpolation approaches were tested with a reduced number of estimation gauges, it was found that the GLMM failed to maintain relatively good precipitation results, and its precipitation RMSE values rose drastically (beyond $100mm$

with 4 gauges only). Once more, this suggests that compared to the alternatives, in this case study the GLMM is more sensitive to the number and distribution of estimation gauges. The importance of the latter is highlighted when using only 14 gauges for model estimation but including at least one of the high elevation gauges, Los Bronces or Lagunitas. This gives an RMSE of $19mm$, which is considerably less than the overall RMSE for the GLMM (32.1 mm).

All other interpolation approaches for precipitation behaved similarly when facing a reduction in the number estimation

gauges. As shown for the LOOCV (see Figure 22), this may be because errors in high elevation gauges strongly influence the overall RMSE. When only 4 gauges were included, however, WCA showed a slightly better RMSE ($23.5mm$) but a larger FAR (27.9%). It was also found that CHIRPS, as a standalone product, represents a useful alternative source of data (i.e. compared to the methods tested in this project, some of which are used in hydrological modelling) when 4 or less gauges were available ($RMSE = 26.2mm$, $POD = 88.5\%$ and $FAR = 28.6\%$).

The results in this paper show how a simple approach that can be easily reproduced elsewhere, may perform at least as well as other more complex or more commonly used approaches, in a catchment with sparse monitoring networks and complex climate dynamics. Based on this evidence and its simplicity, it would be desirable to use WCA to analyse temperature in this case study. For precipitation, WCA is also preferable, unless the modeller was particularly interested in the occurrence of precipitation in the dry season, in which case the GLMM would be desirable if computational requirements are not an issue, or

LR otherwise. Analyses of further case studies are required to be able to generalise these findings.

The fact that 15 temperature gauges in the mountain areas measured during one summer season only, or just started measuring values after 2008, means that although ideal to increase the reliability of results, an analysis of a longer period is still not possible. For precipitation, it would also have been desirable to have good quality gauges between 1300 and 2700 masl, to better understand what happens between the observations in low elevation points and the two high elevation gauges in the

mine sites.

Beyond the issues with the number and location of gauges to estimate the parameters of the GLMM, this paper shows how approximate Bayesian inference methods can be applied to estimate parameters of these models in a hydrological context. Despite there being high computational requirements with the the R-INLA package, these are lower than those of MCMC, and this facilitates the use of GLMMs. It would now be useful to test if the benefits of GLMMs and Bayesian approaches

discussed in this paper and in the non-hydrology literature (Pilz and Spöck, 2008; Ecker and Gelfand, 1997) can equally be





achieved by stochastic approaches like Kriging and GLMs that are more common in hydro-climate applications. It would be particularly interesting to analyse how these approaches behave in well and poorly monitored regions, and how this influences hydrological modelling. Furthermore, it would also be useful to analyse further prior distributions for the Bayesian estimation of the GLMM's parameters. This project used the default priors of the R-INLA package, to facilitate its implementation, but

this could be enhanced.

# 6  Conclusions

Interpolation of climate variables is a major field of research in hydrology, and its relevance is related to the importance of this data for water resources modelling. The scope of this paper was to compare four interpolation approaches for temperature and precipitation in a catchment with complex and steep terrain, and a low density network of gauges.

For temperature, the Generalised Linear Mixed Model (GLMM) reproduced observations in this case study in the best way (i.e. smallest Root Mean Squared Errors - RMSE, in a leave-one-out cross validation - LOOCV), although it was closely followed by a more simple alternative based on merging observations and WorldClim maps (WCA). Inverse Distance Weighting (IDW) and Lapse Rates (LR - i.e. a linear regression using only elevation as a covariate) showed a worse performance.

Furthermore, only WCA demonstrated resilience to the reduction of the number of estimation gauges, and so shows good

prospects for using this alternative to generate input climate data of hydrological models in sparsely monitored catchments. The GLMM, IDW and LR, on the other hand, had larger errors to the point that for this case study, it was desirable to use the long term temperature estimates in the raw WorldClim maps instead, when few gauges were made available.

For precipitation, the LOOCV evidenced that no alternative was clearly superior in this case study in terms of RMSE, and this may be because errors in high elevation points seem to be more determining the overall RMSE, than the quality of

calculations of each approach. All approaches showed a similar resilience to the reduction of estimation gauges, except for the GLMM, which had a poorer performance with a RMSE value larger than $100mm$ when only 4 gauges were made available. WCA showed slightly better RMSE than the rest in most cases, which together with its simplicity, highlights the desirability of the method, even if it did not outperform others as much as for temperature.

In terms of the added value of alternative datasets, it was found that in this case study the inclusion of CHIRPS and World-

Clim maps was as relevant and illustrative as the comparison of interpolation approaches. On the one hand, WorldClim maps allowed developing a very simple but quite efficient method that showed good performance and high resilience when working with few gauges. On the other hand, CHIRPS was demonstrated to be a useful source of precipitation data where few gauges are available. Thus, both represent alternatives to support the development of water resources models in regions with few point observations.

The paper also illustrated how approximate Bayesian inference methods, particularly the INLA-SPDE method available in the R-INLA package, can be used to estimate the parameters of spatio temporal models in hydrological contexts. Further research could explore other spatio-temporal models and prior distributions, and how the results of this approach compare to those of GLMs or Kriging.

**Appendix A: Gauges Used**

*Author contributions.* Juan Ossa-Moreno, Neil McIntyre and Greg Keir conceived the paper. Juan Ossa-Moreno drafted the paper, did the literature review and conducted the research. Greg Keir helped writing the code to run the GLMM. Neil McIntyre advised on hydrological processes and interpolation of climate variables. Michela Cameletti advised on the use of INLA-SPDE, and Diego Rivera advised on the

5    Chilean context, on the interpolation of climate variables in mountain regions and on the literature review. All authors reviewed the paper drafts.

*Competing interests.* There are no competing interests.

*Acknowledgements.* This research was supported by use of the NeCTAR Research Cloud and by QCIF (http://www.qcif.edu.au). The NeC-TAR Research Cloud is a collaborative Australian research platform supported by the National Collaborative Research Infrastructure Strategy.

10   FlashLite was supported by ARC LIEF grant LE140100061, The University of Queensland, Queensland University of Technology, Griffith University, Monash University, University of Technology Sydney, and The Queensland Cyber Infrastructure Foundation. The University of Queensland International Scholarships support the main author. The authors would like to thank the support from James McPhee in Universidad de Chile, Codelco and Anglo-American for providing climate data to the project.





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





**Table A1.** Details of gauges used

| | Station | Elevation | Long | Lat | Variable | Dates available | % of Missing Gaps in the 5 year period |
|---|---|---|---|---|---|---|---|
| 1 | 05200007-6 | 1202 | -70.68 | -32.42 | P | All period | 0 |
| 2 | 05403006-1 | 1313 | -70.36 | -32.92 | P | All period | 1.67 |
| 3 | 05410002-7 | 954 | -70.51 | -32.85 | P | All period | 5 |
| 4 | 05410005-1 | 642 | -70.74 | -32.76 | P | All period | 3.33 |
| 5 | 05410006-K | 1078 | -70.47 | -32.86 | P | All period | 0 |
| 6 | 05410007-8 | 830 | -70.6 | -32.83 | P | All period | 0 |
| 7 | 05410008-6 | 650 | -70.72 | -32.75 | P | All period | 0 |
| 8 | 05414001-0 | 1193 | -70.58 | -32.50 | P | All period | 23.33 |
| 9 | 05414004-5 | 1209 | -70.57 | -32.49 | P | All period | 5 |
| 10 | 05414005-3 | 943 | -70.7 | -32.57 | P | All period | 0 |
| 11 | 05415004-0 | 1023 | -70.6 | -32.68 | P | All period | 1.67 |
| 12 | 05422002-2 | 835 | -70.82 | -32.93 | P | All period | 1.67 |
| 13 | 05732001-K | 575 | -70.8 | -33.09 | P | All period | 1.67 |
| 14 | 05732002-8 | 597 | -70.77 | -33.08 | P | All period | 5 |
| 15 | 05733006-6 | 973 | -70.75 | -32.95 | P | All period | 0 |
| 16 | 05733010-4 | 809 | -70.81 | -32.95 | P | All period | 1.67 |
| 17 | Los Bronces | 3423 | -70.29 | -33.15 | P | All period | 0 |
| 18 | 330019 | 654 | -70.55 | -33.45 | T | All period | 44.58 |
| 19 | 330020 | 529 | -70.68 | -33.45 | T | All period | 0.16 |
| 20 | 330021 | 481 | -70.79 | -33.39 | T | All period | 1.04 |
| 21 | AWS1 | 3088 | -70.11 | -32.99 | T | Summer 08-09 | 96.11 |
| 22 | AWS2 | 2785 | -70.11 | -32.97 | T | Summer 08-09 | 96.11 |
| 23 | AWS3 | 3269 | -70.1 | -33 | T | Summer 08-09 | 96.66 |
| 24 | Angela | 3573 | -70.27 | -33.08 | T | All period | 1.81 |
| 25 | Barroso | 3776 | -70.23 | -33.11 | T | All period | 4.05 |
| 26 | Hornitos | 2214 | -70.15 | -32.87 | T | From Sept/12 | 80.01 |
| 27 | Lagunitas | 2922 | -70.25 | -33.08 | P and T | All period | 0 |
| 28 | MP | 4080 | -70.26 | -33.17 | T | All period | 2.35 |
| 29 | Saladillo | 1585 | -70.28 | -32.93 | T | From Dec/11 | 66.32 |
| 30 | TLog1 | 3254 | -70.1 | -33 | T | Summer 08-09 | 96.22 |
| 31 | TLog10 | 3004 | -70.11 | -32.99 | T | Summer 08-09 | 96.22 |
| 32 | TLog11 | 2968 | -70.11 | -32.98 | T | Summer 08-09 | 96.22 |
| 33 | TLog12 | 2911 | -70.11 | -32.98 | T | Summer 08-09 | 96.22 |
| 34 | TLog2 | 3269 | -70.1 | -33 | T | Summer 08-09 | 96.22 |
| 35 | TLog3 | 3269 | -70.1 | -33 | T | Summer 08-09 | 96.22 |
| 36 | TLog4 | 3212 | -70.1 | -33 | T | Summer 08-09 | 96.22 |
| 37 | TLog5 | 3153 | -70.11 | -32.99 | T | Summer 08-09 | 96.22 |
| 38 | TLog6 | 3081 | -70.11 | -32.99 | T | Summer 08-09 | 96.22 |
| 39 | TLog7 | 3094 | -70.11 | -32.99 | T | Summer 08-09 | 96.22 |
| 40 | TLog8 | 3092 | -70.11 | -32.99 | T | Summer 08-09 | 96.22 |
| 41 | TLog9 | 3070 | -70.11 | -32.99 | T | Summer 08-09 | 96.22 |