# Peer review of "Comparison of approaches to interpolating climate observations in steep terrains with low-density gauging networks"

_Hydrology and Earth System Sciences, 2018_

## Referee Comment (RC1) · Anonymous Referee #1 · 3 Dec 2018

General comments:

The study from Ossa-Moreno, J., et al. aims to compare the performances of four different methods of increasing complexity, from Inverse Distance Weighting (IDW) and Lapse Rate (LR) to the Generalized Linear Mixed Model (GLMM), for spatially interpolating daily/monthly ground station observations of temperature and precipitation from a network with low spatial resolution. The study region is located in the area of the Aconcagua river basin, a mountain catchment located in the Central Andes of Chile. The comparison is performed using a leave-one-out cross-validation technique based on Root Mean Square Error (RMSE), integrated in the case of precipitation by two

other indices, the probability of detection and false alarm ratio. The authors also aim at evaluating the sensitivity of these methods to the number of available ground stations.

I think the topic of the manuscript quite interesting, potentially helping to provide valuable tools based on the integration of different sources of data (e.g., ground stations networks and remote sensed observations) especially in remote mountain areas where station networks are sparse, unevenly distributed, and difficult to maintain.

However, I think that in the present form this manuscript is not suitable for publication and I suggest that a major revision is necessary from the authors, advising them to provide effective responses to the issues here evidenced.

Among all, the present description of the methodology cannot help the reader to correctly interpret the showed results. Core methods are described but there is a general lack of clarity on the type of data in input to each interpolating method. Also, it is not clear which is the final product of these methods (e.g., daily/monthly gridded data on a regular grid of a specific resolution). This information is of particular interest given that the authors stated that the final aim consisted in providing inputs for hydrological and water resources models.

Another important aspect is associated with the sensitivity test with respect to the number of the gauges used for the interpolation. The authors compare the performances of the four interpolating methods, using data from unevenly distributed stations from a network with decreasing spatial resolution from about 1 gauge/400km2 to 1 gauge/4000km2. In this condition, it is not surprising that even a sophisticated (and high computationally demanding) method as the GLMM provides poor performance. Still, without more information on the methodology (see previous paragraph) it is not possible for the reader to interpret the results. This is particularly true since one of the main outcome of this study is that '. . .the WorldClim approach (ed., a combination of IDW with gauge data and WorldClim maps, monthly historical averages obtained by statistical analysis of worldwide weather observations between 1950 and 2000 and

interpolated using latitude, longitude and elevation) may be recommended as being the more accurate, easy to apply and relatively more robust to tested reductions in the number of estimation gauges, particularly for temperature'. On the other hand, even with the lack of information on the methodology, I find very interesting that, besides the study region is a mountain area characterized by a very pronounced topographic gradient, based on the full spatial resolution of the gauges network (∼1 gauge/400km2) almost all methods seem to perform quite well for temperature at daily time resolution. Unsurprisingly, for precipitation, whose character is highly stochastic, the performances of all methods result so poor at daily time resolution that the comparison is performed at monthly time resolution. Concerning this specific aspect of the study, it is not clear why the authors did not apply to the CHIRPS gridded data (Climate Hazards Group InfraRed Precipitation with Station data) a similar approach to the one they used for the WorldClim maps, which would be undoubtedly very interesting especially at the daily time resolution.

Finally, the manuscript is not easy to read, the structure of many sections is confusing, mixing different aspects; more clearly structured sections would be advisable. Number of figures is unusually large, they are not optimized and with a poor layout. Also, most of them could be grouped in multi-panel figures to allow for an easier and faster results comparison.

Specific comments

1. Section 2.1: I would suggest the authors to separate the description of the geographical and climate settings. If the authors are interested in considering in their study the impact of climate variability on the model-parameter estimation, then the climate setting would deserve a more extensive description, including major and relevant literature and clearly provide known impacts on the variability of temperature and precipitation in the region under study. Fig. 1 is not easy to read, redundancy could be reduced by eliminating the actual large map, enlarging the small one and clearly drawing the divide of the portion of the catchment under study. I would suggest changing the color

of the divide being very similar to topography background, therefore difficult to be distinguished. The authors mentioned the importance of glacial/snow melt, a comment on the presence of glaciers in this catchment would be of interest. For example from the Randolph Glacier Inventory [RGI-Consortium, 2017] could be of help.

2. Section 2.2: I would suggest to clearly state at the very beginning the total number of time series and the maximum time span covered by the considered time series. Fig. 2 and 3 could be merged in a 2-panel figure.

3. Section 2.3: I would include references of studies that have evidenced decreased skill of remote sensed products in the mountain environment (lines 27-29). Fig. 4 and 5 could be merged in a 2-panel figure. What is the DEM dataset for? Only for the regressions? Also, the authors could consider including a plot of the MEI index discussing the occurrence of El Nino or La Nina years during the period for which data are available. Given the short length of the used time series, it could be difficult to have enough ENSO cycles to get a significant correlation between the observations and the MEI index.

4. Section 3: in the first paragraph (lines 2-11) it is not clear if the authors refer to literature or to the methods that were used in the present study. I would suggest being clearer and more direct. In fact, it would be useful for the reader to have in this introductory paragraph of the applied methods a structured list of the methods, possibly referring to the literature for advantages/drawbacks. Also, It would be also useful to discuss why the GLMM method (which provides '... larger flexibility to analyze random effects...' than GLMs) is potentially a good tool for interpolating daily temperature and precipitation observation in a complex mountain region. Also, it is not clear which type of data (ground station data or spatial data) are used as inputs for each method, which is quite important information that should be integrated. Finally, it is not clear at which resolution the final interpolated variables (temperature and precipitation) are provided (the same for each method, i.e. the WCA resolution?). The authors are strongly suggested to provide this information.

5. Section 3.1: Within the GLMM method description the reader finds that monthly data and not daily observations (as they were initially defined) are used by the authors for precipitation. This generates confusion, I would suggest to state clearly in the abstract and data description (section 2.2) which is the time resolution for both datasets. If precipitation data have monthly time resolution, it is not clear which added information would bring the test indices POD and FAR.

6. Section 3.2: Is the WCA method based on IDW using both station data and WC map data? Table 1: please consider adding a column for the data used as inputs (station data, spatial data, DEM,. . .). Fig. 6 and 7 could be merged in a 6-panel figure.

7. Section 3.3: I suggest the authors to clearly divide the two comparison tests, LOOCV and sensitivity and avoid mixing the two tests. The authors also add that the RMSE estimation for the GLMM method was performed using the expected values of each variable for each time step (line 18). How it is calculated for the other methods? Should not be the same? That is, for all the methods, do not the authors generate time series of gridded data that are compared then with the station values with the LOOCV method? This aspect should be clarified; otherwise it is very difficult to correctly interpret the results in Table 3 and 4. In fact, it is not clear how the authors could obtain a so small RMSE for the raw WorldClim maps values (monthly worldwide estimated averages compared with daily station data?).

8. Section 4: In the results section, the authors provide correlation values but they do not explain the purpose of this part of analysis. It seems quite an important pre-processing step that aims at testing which variates are to be used for parametrizing the relevant interpolating schemes. I suggest the authors adding a paragraph in the methods section that explains this aspect. Fig. 8 to 13 could be merged in a 6-panel figure.

9. Section 4.1: How did the authors calculate the daily temperature averaged over the 5-y period? Is i-th daily value the simple mean of all days i (i.e., average of 5 values)?

Or did the authors consider a moving window k-days long (i.e., average of 5*k values)? I would suggest clarifying this and add a sentence describing how the data in Fig. 14 to 16 have been estimated including smoothing method description. Furthermore, it would be also interesting to provide a comment on why we look at the daily temperature averaged over the 5-y period, i.e., which added information this comparison provides. Finally, Fig. 14 to 16 could be merged in a 3-panel figure.

10. Section 4.2: Why the CHIRP data have not been used in association with the station data? RMSE comparison with WCA indicates a better performance of CHIRPS data in the raw configuration, therefore it is expected that they would much better perform in combination with station data than the rest of the methods. Consider merging Fig. 17 and Fig. 23 (maybe scatter plots are easier to be looked at), Fig. 19 to 21, Fig. 18 and Fig. 22.

---

## Referee Comment (RC2) · Anonymous Referee #2 · 16 Jan 2019

General comments: I think the topic of the manuscript is interesting and also in the scope of HESS, as it intends to answer the relevant question: "which data sets and (interpolation) methods are most adequate to represent climatic conditions and information at altitudes higher than 1000m in data scarce Chile and similar Andean regions." However, in my opinion, the manuscript in its current version is not suitable for publication and needs major revision in many aspects. The following sections need to be improved: Introduction: How have other authors addressed this topic? There is a strong discourse on this issue and a large number of researchers developing precipitation products as MSWEP, CHIRPS and CR2 have dealt with this problem. Please elaborate on the findings of other authors working with high elevation data. Also how

do authors deal with missing information in hydrological modelling, which interpolation methods have worked and which were the results of evaluating different satellite based and combined precipitation data sets in data scarce Andean regions? Although you mention some authors, their findings are not described or compared. Ideally, these should help to justify your objectives. Data: - The data (input, validation..) should be presented in the main text. Otherwise the numbers in the map are useless. Also in the map, it would help to enlarge it and use other colours for elevation and delineate a stronger catchment area to make the map understandable even in black and white. Numbers in the map should also be visible in Figures 2 and 3. - It is not well explained why you only used such a short period. There are enough data available to fill gaps (CR2 P dataset, Chirps, MSWEPv2.2, etc.). Temperature of course is difficult but at least different time periods could be compared. The main variable of interest should be precipitation. - Why do you present a spatial distribution of Chirps in May 2009 instead of comparing it with values from observed data? Methods section 3: - The first paragraphs of this section should be part of the introduction as they deal with the general state of the art. - The advantage of using GLMMs and its exact output in this context is not clear to me. - There should be a conceptual figure explaining the methodology, input data and outputs - You use station data and as Covariates Chirps and ENSO as model input to test different interpolation methods. Then in the results section you correlate station data with Chirps and other data products for the station pixel? This part should be shifted to the data section and justify the method and data input (or not?). - 4.1 difference between input data and validation data not presented Results: In light of the above described missing information regarding the data input, validation data and output variables, it is difficult to understand the results and their interpretation. Overall presentation structure and language are still very poor. There are too many figures with little information content. Please focus on the main findings and try to present them in fewer self-explanatory figures.
* * *
505, 2018.

---

## Author Comment (AC1) · 11 Feb 2019

Comparison of approaches to interpolating climate observations in steep terrains with low-density gauging networks Response to comments by Reviewers We are very grateful with the two anonymous reviewers who have provided very valuable feedback to improve the manuscript. We are glad that both of them highlighted that the topic of the manuscript is interesting, valuable and within the scope of HESS. We are also happy that reviewer 1 highlighted the value of the temperature results.

Overall, the key requirements from reviewers involve: reorganising the content of some sections, better explaining the results obtained by the referenced authors on the in-

terpolation of climate variables in mountain areas, providing more information of the general climate in the case study and giving a better explanation of the main method used to interpolate the variables. Furthermore, we will do the extra analysis suggested by reviewer 1 in his last comment, to have a better idea of how the WCA method could be applied to the CHIRPS dataset. We address all the comments of the reviewers below. We are grateful for comments received, as they will improve considerably the quality of the paper, however, since none of the comments involves major changes, we are confident we will be able to fully address all of them and re-submit the manuscript within two months. We will present as follows the corrections that have been done up to date, and the plan to address the rest.

Reviewer 1 Specific Comments

1. Section 2.1 - Separate the description of the geographical and climate settings.

The description of the geography was moved to the beginning of Section 2, while the climate settings were kept in Section 2.1.

a. Climate setting would deserve a more extensive description

The description of the climate settings in Section 2.1 was considerably increased, to provide more details of the broader climate phenomena affecting the case study, the sources of inter-annual variability (including ENSO and a brief comment on the Pacific Decadal Oscillation), and temperature fluctuations. Further references were included. A comment on glaciers was also added, including two references which provide further details. We did not go deeper on the subject of glaciers, as their presence is restricted to the highest elevation areas of some of the sub-catchments in the case study. Furthermore, one of the references highlights that, although challenging to quantify, their role in catchment flows seems to be only relevant during dry years, and only for the very upper sub-catchments (Ohlanders et al., 2013). This means that the overall relevance of glaciers in the case study is not that high, thus, we do not consider pertinent to provide much more detail about them.

b. Eliminate large map from Figure 1. Enlarge the small one. Clearly define the case study. Change colour of catchment delineation.

The catchment delineation colour has been changed so it is easy to identify it. The whole figure has been enlarged. None of the figures was eliminated as we think they all are useful to clearly locate the catchment.

c. Include comments on glaciers in the area.

See 1a.

2. Section 2.2 – Clearly state the total number of time-series and the maximum time-span covered by the considered time-series.

This information was provided in the Appendix, however, we acknowledge that a better explanation was required in the text. Thus, the third paragraph of Section 2.2 was reworded to better link the text with the information provided in the Appendix. This paragraph is included as follows: "A total of 41 gauges were used in the project, 17 of them measured precipitation only, 23 measured temperature only and 1 measured both variables. The location of the temperature and precipitation gauges is shown in Figure 1, while further details of the gauges (including the periods with information available and the percentage of missing values) are provided in Table A1 in the Appendix."

a. Merge Figure 2 and 3 in a two panel figure.

We acknowledge that the large number of figures was an issue in the previous version. This was caused by the fact that our original latex text was not in Copernicus format and included several files for the same figure (i.e. the files of figures 2 and 3 were merged in latex into one figure). However, having several files for the same figure is not allowed in Copernicus format, thus we separated all files into different figures, without realising the negative consequences in the paper All files that were part of the same figure were merged in R, as to create one file only, and in this way there is only one figure, for the previous figure 2 and figure 3. The large number of figures was an

issue in the previous version. Each figure that we planned to be a multiplot had to be split to follow the Copernicus Latex format, they were presented as multiple separate figures. Now, image files were merged prior to their inclusion in Latex so that many figures are now merged appropriately, including figure 2 and figure 3.

3. Section 2.3 – Lines 27-29 include references of studies that have evidenced decreased skill of remote sensed products in the mountain environment.

The references were already included in the previous sentence (Dinku et al., 2010, Manz et al., 2016, Thiemig et al., 2012). However, they were repeated to make it more explicit that they provide the evidence of the decreased skill of remote sensed products to work in mountain areas or with extreme weather conditions, when compared to flat regions. The new paragraph is presented as follows: "To complement the point observations, the Climate Hazards Group InfraRed Precipitation with Station data (CHIRPS) satellite product (Funk et al., 2015) was used. Including remotely sensed data to analyse climate variables is increasingly popular amongst researchers, and several examples exist for precipitation in the Andes (Dinku et al., 2010, Zambrano-Bigiarini et al., 2016, Manz et al., 2016, Álvarez Villa et al., 2011) and beyond (Nikolopoulos et al., 2013, Thiemig et al., 2012, Dinku et al., 2014). Based on these experiences in mountain regions, it could be said that generally, satellite products tend to be good at detecting precipitation and its overall spatial variability, but struggle to predict the magnitudes of the events, particularly heavy precipitation events, and for daily and subdaily resolutions (Dinku et al., 2010, Manz et al., 2016, Thiemig et al., 2012). This is usually a consequence of orographic effects and convective precipitation events."

a. Merge Figure 4 and 5 in a 2 panel figure.

Figures have been merged. See answer to comment 2a.

b. Specify what is the DEM used for.

The DEM was used to define the elevation at all points in the catchment, as this variable

is required for some of the interpolation approaches. The first part of this paragraph in section 2.3 was adjusted to include this as follows: "The third spatial data set used was a Digital Elevation Model (DEM) based on the Shuttle Radar Topography Mission (SRTM) (Jarvis et al., 2008), with a spatial resolution of 90 m. The DEM was used to define the elevation, which is an input variable in some of the interpolation approaches."

c. Consider including a plot of MEI index to discuss el nino or la nina events during the period of analysis. Given the short length of the used time series, it could be difficult to have enough ENSO cycles to get a significant correlation between observations and MEI index.

We considered this and concluded that there would not be too much added value by including this plot. As the reviewer highlights, the period of analysis is relatively short compared to the frequency of the ENSO events, and this may have hindered finding a better correlation between the MEI index and the climate variables. This, however, is clearly stated in section 4, where we describe the correlation analysis between variables and covariates.

4. Section 3 – It is not clear if in the first paragraph the authors discuss literature or the methods

We have moved most of the content in the beginning of Section 3 to the introduction, and kept only one paragraph explaining the reason for using the GLMM in this project.

a. Include at the beginning a structured list of the methods, possibly including literature on advantages/drawbacks.

We will include a list (i.e. perhaps as a table) at the beginning of this section, providing references, and potential advantages and disadvantages of all methods used.

b. Discuss in more detail why the GLMM is potentially good for this application.

We consider this is one of the key comments from reviewers, and we will make sure to provide a more in-depth explanation of the GLMM and on the added value of this

method, compared to alternatives in hydrology. We think we did not overlook this, but in the original manuscript we focused more on providing details of the mathematics (Section 3.1), rather than on the description of the benefits.

c. Be more clear on what type of data was used for each method.

We thought sufficient details of each source of data was provided in Section 2, however, we will make sure to clarify the specific data requirements of each method (i.e. what data was used on each method).

d. Provide details of the resolution of the climate outputs.

This will be provided at the end of section 3.

5. Section 3.1 – Clearly state in Section 2.2 and abstract that monthly precipitation data was used.

Done

a. If monthly precipitation data was used, why including FAR and POD?

Although POD and FAR are more commonly used for daily analyses, the large numbers of months without precipitation in the catchment make the calculation of these two categorical statistics valuable. This reason was made explicit in the article with the following paragraph: "Furthermore, two categorical statistics, the False Alarm Ratio (FAR) and the Probability of Detection (POD), were used to assess to what extent the model is able to predict precipitation occurrence (see Table 2)(Zambrano-Bigiarini et al., 2016). The latter is relevant, even at a monthly time-scale, taking into account that in the case study, there are several months without any precipitation (above the defined significance threshold), thus properly simulating its occurrence is not a trivial exercise."

6. Section 3.2 – Is WCA based on IDW using both station data and WorldClim maps?

Yes, WCA is based on using IDW to interpolate the residuals between the WorldClim maps and the station data. We thought this was clear enough; however, we reworded

the explanation to further clarify. Two references explaining a similar method were included in the revised version, in case the reader wants to have more details about this procedure. "The WCA method attempts to couple the benefits of the spatial variability of the WC maps and those of the temporal resolution of the observations in a simple way. This approach is similar to the RIDW in (Manz et al., 2016) or the bias adjustment in (Dinku et al., 2014), but in this case using WC maps. The residual between observations and WC maps is computed at each gauge location, these residuals are interpolated using Inverse Distance Weighting (IDW) to each point in the catchment, and this interpolated surface is added back to the original WC map. This procedure is repeated for every time-step (monthly for precipitation and daily for temperature)."

a. Merge figures 6 and 7

To address the previous comment we reviewed again some papers where similar methods were applied, and realised that none of them included this kind of figures, but only a brief explanation with the steps followed. Taking this into account, figure 6 and 7 were eliminated and the explanation of the method was improved by providing a more specific explanation, and some references to obtain further details about the approach.

7. Section 3.3 – Divide LOOCV and sensitivity tests.

The explanation of the LOOCV and the sensitivity test was divided. The first paragraph in Section 3.3 explains the LOOCV while the next two explain the sensitivity tests. LOOCV and sensitivity results in section 4 will also be separated.

a. Be more clear why for the GLMM it was required to use the expected value as opposed to the others (GLMM is a stochastic method, the others are not). Explain this in a better way for all methods.

Further details of this were provided in the fourth paragraph of Section 3.3, as follows: For all tests, the average Root Mean Squared Error (RMSE) of the validation group was used to assess the performance of temperature and precipitation predictions, following

similar comparisons (Cameletti et al., 2013, Manz et al., 2016, Nerini et al., 2015). Being a stochastic method, for the GLMM this involved the analysis of the expected values of each variable for each time-step (y in Equations 5 and yP in Equation 8). On the other hand, the other three methods are deterministic, thus the single set of values at each time-step (e.g. y in Equation 11) were used for the RMSE computations.

b. Be more specific of the comparisons of raw WC maps and temperature data, and discuss its small RMSE.

We thought sufficient details had been provided, however, we will make sure to discuss in more detail the small RMSE for raw WC maps. Furthermore, with the changes in the explanation of the WCA method (See comments 6 and 6a), we hope it is more clear how the comparison between WC maps and temperature data was done.

8. Include a paragraph in the methods section discussing the correlation analysis.

A paragraph has been included at the end of Section 3.1, briefly describing the correlation analysis and its purpose. The more in-depth discussion of the results was kept in the first paragraph of the results section (Section 4). The paragraph included is as follows: "Furthermore, before including the covariate data in the GLMM (e.g. WC, elevation, CHIRPS), an analysis of their correlation with the climate variables was done. This included plotting temperature and precipitation observations versus the covariates, and computing Pearson Correlation coefficients. This analysis was used to define what covariates to include in each GLMM."

a. Merge Fig 8-13 in a 6 panel figure.

The figures have been merged. See answer to comment 2a.

9. Section 4.1 – Explain how and why the 5 yrs daily average was calculated, and explain that this was for plotting purposes only in Figs 14-16.

This aggregation was done for illustration purposes only. Our goal with these figures was to show: what methods over and under-estimate observations, by approximately

how much, how this changed as a function of the period of the year, and how this changed as a function of different types of stations. The 5-year series of daily data contained too much variability to visually assess the trends, which was achieved using the averaged series. For the same reason, to facilitate the visualisation of the main trends, values were also smoothed using the LOESS method. Briefly, the method analyses data nearby a point X (how much data is included is a user defined parameter), and does a simple regression using this data. The value of X is adjusted to the value predicted by this regression. Although this may eliminate day-to-day fluctuations, the overall trend over several days is shown much more clearly, as the noise is reduced. The LOESS is just one of the several methods that could be used to do this (a simple moving average could have also been used). A reference was provided so the reader can have access to more details (Jacoby, 2000). This information was not provided in the previous version because we did not consider it to be very relevant, taking into account that the method is only used for illustration purposes. We will provide more detail about the purpose of this aggregation and the method in the text.

a. Merge these three in a three panel figure.

The figures have been merged. See answer to comment 2a.

10. Why CHIRPS data was not analysed in the same way as WC? Or be more clear how the CHIRPS data was merged with observations.

Chirps was not applied in the same way as WC because Chirps does not include temperature data. However, it is possible to apply Chirps in the same way as WC to interpolate precipitation data. We understood the comment from the reviewer and realised that it would be valuable to show how CHIRPS would behave with this method, taking into account its better performance than WC. This will included in the revised manuscript.

a. Merge figures 17 and 23. 19 and 21. 18 and 22.

The figures have been merged. See answer to comment 2a.

Reviewer 2 Specific Comments Introduction

How have other authors addressed this topic? There is a strong discourse on this issue and a large number of researchers developing precipitation products as MSWEP, CHIRPS and CR2 have dealt with this problem. Please elaborate on the findings of other authors working with high elevation data. Also how do authors deal with missing information in hydrological modelling, which interpolation methods have worked and which were the results of evaluating different satellite based and combined precipitation data sets in data scarce Andean regions? Although you mention some authors, their findings are not described or compared. Ideally, these should help to justify your objectives.

We did an extensive review of similar analyses, both in terms of interpolation techniques and merging of satellite data with observations, and many of these references were included in the original paper. We acknowledge, however, that the findings of these authors could be described in further detail in the introduction section. Based on this we can conclude what the gaps in knowledge are, in order to justify the objectives of this paper.

Data The data (input, validation..) should be presented in the main text. Otherwise the numbers in the map are useless. Also in the map, it would help to enlarge it and use other colours for elevation and delineate a stronger catchment area to make the map understandable even in black and white. Numbers in the map should also be visible in Figures 2 and 3.

We are not sure about what the reviewer means by including the "data (input,validation..)" in the main text. We have tried to follow general practice from similar papers working with similar data, which commonly include: • A map of the region being analysed including the location of the gauges. • A list, usually in an appendix, of the stations analysed, providing detailed information of the location, variables mea-
sured and availability of observations (this is not included when analyses involve a very large number of stations e.g. > 100). • General statistics of the stations (e.g. mean and range), plus some figures of some stations or from a region, describing general trends of the data (e.g. seasonality). Furthermore, we are not sure how we could differentiate validation stations, as a leave-one-out cross validation method was used, which means that all stations were both used for calibration and validation in different runs of the model. The map has been updated following the comments from both reviewers, to make sure that the catchment is easy to identify, terrain elevation is easy to differentiate and the location of the stations is clearer. Figures 2 and 3 were updated as well following comments from reviewers. Also, a CHIRPS figure was provided for the reader to visualise this product and compare it with the WC data, particularly the resolution of both within the area of analysis. We did not think about comparing it with observations as did not see the purpose of a single month comparison, however, we will include the average values of CHIRPS and WC in Figure 2 to facilitate a more robust comparison of data.

It is not well explained why you only used such a short period. There are enough data available to fill gaps (CR2 P dataset, Chirps, MSWEPv2.2, etc.). Temperature of course is difficult but at least different time periods could be compared. The main variable of interest should be precipitation. - Why do you present a spatial distribution of Chirps in May 2009 instead of comparing it with values from observed data?

We were interested in analysing both temperature and precipitation in the catchment over the same period. We obtained 5 years of valuable data, previously not used for research, from a company operating in the area, who installed multiple weather stations around 2008, thus we started our analysis in that year. We acknowledge that there are precipitation stations in the lowlands (government gauges which we obtained from CR2) and one in the mountains, which have been continuously measuring precipitation for decades. However, the availability of multiple gauges in the mountains for the 5 year period starting on 2008 was decisive in terms of the quality of the research. The temporal infilling using these products could have been inappropriate for creating a data set for assessing the spatial interpolation methods. We thought that the spatial interpolation methods would be better assessed using observations from gauges. The use of alternative spatial data sets is useful when used as inputs to the spatial interpolation methods, which he have done, as long as available observations from gauges are used to assess their proficiency. We used datasets such as the ones that the reviewer suggests, e.g. we used CHIRPS and WC maps, however, the scope of the paper was not to use all of the data sets available for the case study but to analyse the interpolation approaches. We would like to further stress that we have never attempted to claim that the results in the paper are valid for long term trends, nor that the conclusions are valid under all circumstances. We have been cautious highlighting that our findings are restricted by the limitations of the study, however, this does not mean that they are not useful. We think that they provide valuable information of the performance of some methods, under a complex climatic region with few observation gauges. We will explain all of this in much more detail in all sections of the revised manuscript.

Methods section 3: The first paragraphs of this section should be part of the introduction as they deal with the general state of the art. - The advantage of using GLMMs and its exact output in this context is not clear to me. - There should be a conceptual figure explaining the methodology, input data and outputs - You use station data and as Covariates Chirps and ENSO as model input to test different interpolation methods. Then in the results section you correlate station data with Chirps and other data products for the station pixel? This part should be shifted to the data section and justify the method and data input (or not?). - 4.1 difference between input data and validation data not presented.

Following the comments from both reviewers, we have moved some of the information from the methods section to introduction, and we will describe the advantages of the GLMM in a much better way. Furthermore, we will better describe this method, its

inputs and outputs in the list of methods we will include in section (see answer 4a to the comments from the first reviewer). We will make sure to explain here what information was used as covariates/input for each method.

As suggested by reviewer 1, we included a new paragraph in Section 3 better describing the correlation analysis between climate variables and covariates. However, we consider that it is better to keep the outcomes of this analysis in the results section, as they are part of the process to build the GLM (i.e. defining the covariates to use).

We are not sure what the reviewer means by differences between input data and validation data. By using a leave-on-out cross validation (Manz et al., 2016), we believe we go a step ahead of using one part of the data for estimation purposes, and the rest for validation. We run each method several times, and in each of them we remove one station at a time, to validate the results of that specific run. We repeat this process for all stations, which means that all stations were used for estimation purposes, but at the same time each of them was used once for validation purposes. The overall output is the average results of all validation stations (i.e. all stations, but only when they were used for validation). Perhaps we are not very clear with this, thus we will make sure to provide a better description in the revised manuscript.

Results: In light of the above described missing information regarding the data input, validation data and output variables, it is difficult to understand the results and their interpretation. Overall presentation structure and language are still very poor. There are too many figures with little information content. Please focus on the main findings and try to present them in fewer self-explanatory figures.

Once more, we are not clear what the reviewer means by issues with data input and validation data (see previous answer), but we will make sure to better explain the leave-one-out cross validation in the revised manuscript. If this was more related to the fact that it is not clear what sets of inputs/covariates were used in each method, we will make sure to clarify this as well in Section 3.

We have improved the structure of the manuscript taking into account the comments from reviewers, particularly in the introduction, data and methods section. We will make sure to double check potential language issues.

The large number of figures was an issue in the previous version and we acknowledge this decreased the presentation quality of that version. As explained in the answer to the comment 2A of reviewer 1, we have solved this issue by merging lots of figures in multi-plots.

For the revised version, we will make sure to focus on the valuable outcomes of the paper and present them in a clearer way.

References

ÁLVAREZ VILLA, O. D., VÉLEZ, J. I. & POVEDA, G. 2011. Improved long‐term mean annual rainfall fields for Colombia. International Journal of Climatology, 31, 2194-2212. CAMELETTI, M., LINDGREN, F., SIMPSON, D. & RUE, H. 2013. Spatio-temporal modeling of particulate matter concentration through the SPDE approach. AStA Advances in Statistical Analysis, 97, 109-131. DINKU, T., HAILEMARIAM, K., MAIDMENT, R., TARNAVSKY, E. & CONNOR, S. 2014. Combined use of satellite estimates and rain gauge observations to generate high‐quality historical rainfall time series over Ethiopia. International Journal of Climatology, 34, 2489-2504. DINKU, T., RUIZ, F., CONNOR, S. & CECCATO, P. 2010. Validation and Intercomparison of Satellite Rainfall Estimates over Colombia. Journal of Applied Meteorology and Climatology, 49. FUNK, C., PETERSON, P., LANDSFELD, M., PEDREROS, D., VERDIN, J., SHUKLA, S., HUSAK, G., ROWLAND, J., HARRISON, L. & HOELL, A. 2015. The climate hazards infrared precipitation with stations–a new environmental record for monitoring extremes. Scientific data, 2. JACOBY, W. G. 2000. Loess:: a nonparametric, graphical tool for depicting relationships between variables. Electoral Studies, 19, 577-613. JARVIS, A., REUTER, H. I., NELSON, A. & GUEVARA, E. 2008. Hole-filled SRTM for the globe Version 4, available from the CGIAR-CSI SRTM

90m Database. MANZ, B., BUYTAERT, W., ZULKAFLI, Z., LAVADO, W., WILLEMS, B., ROBLES, L. A. & RODRIGUEZ‐SANCHEZ, J. P. 2016. High‐resolution satellite‐gauge merged precipitation climatologies of the Tropical Andes. Journal of Geophysical Research: Atmospheres, 121, 1190-1207. NERINI, D., ZULKAFLI, Z., WANG, L.-P., ONOF, C., BUYTAERT, W., LAVADO, W. & GUYOT, J.-L. 2015. A comparative analysis of TRMM-rain gauge data merging techniques at the daily time scale for distributed rainfall-runoff modelling applications. Journal of Hydrometeorology. NIKOLOPOULOS, E. I., ANAGNOSTOU, E. N. & BORGA, M. 2013. Using high-resolution satellite rainfall products to simulate a major flash flood event in northern Italy. Journal of Hydrometeorology, 14, 171-185 %@ 1525-755X. OHLANDERS, N., RODRIGUEZ, M. & MC PHEE TORRES, J. 2013. Stable water isotope variation in a Central Andean watershed dominated by glacier and snowmelt. Hydrology and Earth System Sciences, 9. THIEMIG, V., ROJAS, R., ZAMBRANO-BIGIARINI, M., LEVIZZANI, V. & DE ROO, A. 2012. Validation of satellite-based precipitation products over sparsely gauged African river basins. Journal of Hydrometeorology, 13, 1760-1783. ZAMBRANO-BIGIARINI, M., NAUDITT, A., BIRKEL, C., VERBIST, K. & RIBBE, L. 2016. Temporal and spatial evaluation of satellite-based rainfall estimates across the complex topographical and climatic gradients of Chile. Hydrology and Earth System Sciences.

Please also note the supplement to this comment:
https://www.hydrol-earth-syst-sci-discuss.net/hess-2018-505/hess-2018-505-AC1-supplement.pdf

---

## Author Response (AR1)

**Comparison of approaches to interpolating climate observations in steep terrains with low-density gauging networks**

**Response to comments by Reviewers**

We are very grateful with the two anonymous reviewers who have provided very valuable feedback
to improve the manuscript. We are glad that both of them highlighted that the topic of the manuscript
is interesting, valuable and within the scope of HESS. We are also happy that reviewer 1 highlighted
the value of the temperature results, and that he or she suggested to analyse CHIRPS data in a similar
way as WCA. This new method ended up being a very good alternative to interpolate precipitation.

Overall, the key requirements from reviewers involved: reorganising the content of some sections,
better explaining the results obtained by the referenced authors on the interpolation of climate
variables in mountain areas, providing more information of the general climate in the case study and
giving a better explanation of the GLMM.

We addressed all the comments of the reviewers below.

**Reviewer 1**

**Specific Comments**

1.  **Section 2.1 - Separate the description of the geographical and climate settings.**

*The description of the geography was moved to the beginning of Section 2, while the climate settings
were kept in Section 2.1.*

a.  **Climate setting would deserve a more extensive description**

*The description of the climate settings in Section 2.1 was considerably increased, to provide more
details of the broader climate phenomena affecting the case study, the sources of inter-annual
variability (including ENSO and a brief comment on the Pacific Decadal Oscillation), and temperature
fluctuations.*

*Further references were included. A comment on glaciers was also added, including two references
which provide further details. We did not go deeper on the subject of glaciers, as their presence is
restricted to the highest elevation areas of some of the sub-catchments in the case study. Furthermore,
one of the references highlights that, although challenging to quantify, their role in catchment flows
seems to be only relevant during dry years, and only for the very upper sub-catchments (Ohlanders et
al., 2013). This means that the overall relevance of glaciers in the case study is not that high, thus, we
do not consider pertinent to provide much more detail about them.*

b.  **Eliminate large map from Figure 1. Enlarge the small one. Clearly define the case
study. Change colour of catchment delineation.**

*The catchment delineation colour has been changed so it is easy to identify it. The whole figure has
been enlarged. None of the figures was eliminated as we think they all are useful to clearly locate the
catchment.*

c.  **Include comments on glaciers in the area.**

*See 1a.*

40    **2.  Section 2.2 – Clearly state the total number of time-series and the maximum time-span covered by the considered time-series.**

*This information was provided in the Appendix, however, we acknowledge that a better explanation was required in the text. Thus, the third paragraph of Section 2.2 was reworded to better link the text with the information provided in the Appendix. This paragraph is included as follows:*

45    *"A total of 42 gauges were used in the project, 18 of them measured precipitation and 24 measured temperature. The 42 gauges covered 41 sites, with one site (site 27) having both temperature and precipitation gauges. The locations of the temperature and precipitation gauges are shown in Figure 1, while further details of the gauges (including the periods with information available and the percentage of missing values) are provided in Table A1 in the Appendix."*

50    *In addition, throughout the text, whenever we referenced a gauge, we changed or complemented their names with the number of the site in Figure 1, so it is easier for the reader to locate the gauges in the map without going to the Appendix.*

   ***a.  Merge Figure 2 and 3 in a two panel figure.***

*The large number of figures was an issue in the previous version. Each figure that we planned to be a*
55    *multiplot had to be split to follow the Copernicus Latex format, they were presented as multiple separate figures. Now, image files were merged prior to their inclusion in Latex so that many figures are now merged appropriately, including figure 2 and figure 3.*

*We have also done a major review on the figure and their labels, and have improved their overall presentation.*

60    **3.  Section 2.3 – Lines 27-29 include references  of studies that have evidenced decreased  skill of remote sensed products in the mountain environment.**

*Taking into account the suggestions from Reviwer 2 to reorganise the introduction, we moved this information to the introduction. The new paragraph includes the references supporting each one of the statements.*

65    *"A broader review of the performance of satellite products for estimating precipitation in the Andes and other mountain areas (Nikolopoulos et al., 2013, Thiemig et al., 2012, Dinku et al., 2014), suggests that in these regions, satellite products tend to be good at detecting precipitation (except in very dry areas (Zambrano-Bigiarini et al., 2016, Manz et al., 2016)) and its overall spatial variability, but struggle to accurately predict the magnitudes of the events, particularly during extremely dry (e.g. in*
70    *the north of Chile (Zambrano-Bigiarini et al.)) or extremely wet regions (e.g. western slopes in the Colombian Andes (Dinku et al., 2010)), and for daily and subdaily resolutions (Dinku et al., 2010, Manz et al., 2016, Thiemig et al., 2012)."*

   a.  **Merge Figure 4 and 5 in a 2 panel figure.**

*Figures have been merged. See answer to comment 2a.*

75    **b.  Specify what is the DEM used for.**

*The DEM was used to define the elevation at all points in the catchment, as this variable is required for some of the interpolation approaches. The first part of this paragraph in section 2.3 was adjusted to include this as follows:*

*"The third spatial data set used was a Digital Elevation Model (DEM) based on the Shuttle Radar Topography Mission (SRTM) (Jarvis et al., 2008), with a spatial resolution of 90 m. The DEM was used to define the elevation in the catchment, in order to use this variable in some of the interpolation approaches."*

    *c.* **Consider including a plot of MEI index to discuss el nino or la nina events during the period of analysis. Given the short length of the used time series, it could be difficult to have enough ENSO cycles to get a significant correlation between observations and MEI index.**

*We considered this and concluded that there would not be too much added value by including this plot. As the reviewer highlights, the period of analysis is relatively short compared to the frequency of the ENSO events, and this may have hindered finding a better correlation between the MEI index and the climate variables. This, however, is clearly stated in section 4, where we describe the correlation analysis between variables and covariates.*

    *4.* **Section 3 – It is not clear if in the first paragraph the authors discuss literature or the methods**

*We have done relevant changes in Section 3 and the introduction. We moved all the review of literature to the introduction, and in Section 3 kept only the detailed explanation of the added value of the GLMM within the context of this case study, how our method differs from alternatives like Kriging or GLMs and why we chose an approximate Bayesian inference method.*

    a. **Include at the beginning a structured list of the methods, possibly including literature on advantages/drawbacks.**

*This was included at the beginning of Section 3 in Table 1.*

    b. **Discuss in more detail why the GLMM is potentially good for this application.**

*We provide a detailed explanation of this at the beginning of Section 3.1.*

    c. **Be more clear on what type of data was used for each method.**

*This was clearly included in Table 1.*

    d. **Provide details of the resolution of the climate outputs.**

*The GLMM, IDW, LR and ChA methods were used to generate data at centre-points of 5 km x 5km grids, while WCA used the original 1 km x 1 km WC grids. However, we thought it was more important to specify in the paper how we generated the data for comparison with the validation gauges. This is explained in Section 3.3, where we specify that we generated the estimates at the location of the validation gauges at each round of the leave-one-out cross validation, and then we repeated this for each approach, both for temperature and precipitation.*

    *5.* **Section 3.1 – Clearly state in Section 2.2 and abstract that monthly precipitation data was used.**

*Done*

*a.* **If monthly precipitation data was used, why including FAR and POD?**

*Although POD and FAR are more commonly used for daily analyses, the large numbers of months without precipitation in the catchment make the calculation of these two categorical statistics valuable. This reason was made explicit in the article with the following paragraph:*

*"Furthermore, two categorical statistics, the False Alarm Ratio (FAR) and the Probability of Detection (POD) (e.g. as applied in (Zambrano-Bigiarini et al., 2016)), were used to assess to what extent the model is able to predict precipitation occurrence (see Table 2). These categorical statistics are relevant, even at a monthly time-scale, considering that in the case study there are several months without any precipitation, thus accurately simulating its occurrence is not a trivial exercise."*

*6.* **Section 3.2 – Is WCA based on IDW using both station data and WorldClim maps?**

*Yes, WCA is based on using IDW to interpolate the residuals between the WorldClim maps and the station data. We thought this was clear enough; however, we reworded the explanation in Table 1 and in Section 3.2 to further clarify. Two references explaining a similar method were included in the revised version, in case the reader wants to have more details about this procedure. Taking into account comment 10 from Reviewer 1, here we also explained how the same method was applied with CHIRPS data.*

*"The WCA method attempts to couple the benefits of the spatial variability of the WC maps and those of the temporal resolution of the observations in a simple way. Likewise, ChA attempts to improve the performance of raw CHIRPS by doing a straightforward merging of this product with observations. These approaches are similar to the RIDW in Manz et al. (2016) or the bias adjustment in Dinku et al. (2014), but in this case using WC maps and CHIRPS. First, the residual between observations and WC/CHIRPS is computed at each gauge location at a daily resolution for temperature and at a monthly resolution for precipitation. Then, these residuals are interpolated using Inverse Distance Weighting (IDW) to each point in the catchment, and this interpolated surface is added back to the original WC/CHIRPS values. This procedure is repeated for every time-step."*

*a.* **Merge figures 6 and 7**

*To address the previous comment we reviewed again some papers where similar methods were applied, and realised that none of them included this kind of figures, but only a brief explanation with the steps followed. Taking this into account, figure 6 and 7 were eliminated and the explanation of the method was improved by providing a more specific explanation, and some references to obtain further details about the approach.*

*7.* **Section 3.3 – Divide LOOCV and sensitivity tests.**

*The explanation of the LOOCV and the sensitivity test was divided. Section 3.3 explains the LOOCV while Section 3.4 explain the sensitivity tests. The results were also divided in Section 4. First, a table with the results of the LOOCV is provided, together with its explanation, and then another table with the results of the sensitivity analysis is included.*

*a.* **Be more clear why for the GLMM it was required to use the expected value as opposed to the others (GLMM is a stochastic method, the others are not). Explain this in a better way for all methods.**

*Further details of this were provided in the fourth paragraph of Section 3.3, as follows:*

*"For all tests, the average Root Mean Squared Error (RMSE) was used to assess the performance of temperature and precipitation predictions, following similar comparisons (Cameletti et al., 2013, Manz et al., 2016, Nerini et al., 2015). Being a stochastic approach, for the GLMM this involved the analysis of the expected values of each variable (y in Equation 3 and $y^p$ in Equation 6).*

**b. Be more specific of the comparisons of raw WC maps and temperature data, and discuss its small RMSE.**

*We thought sufficient details had been provided, however, we have improved the description of the comparison of this in Section 3.4.*

*"The sensitivity test was complemented with the estimation of precipitation and temperature values at all locations using raw WC maps and CHIRPS, in order to understand the accuracy of these data sets when used independently of the observations. This involved comparing the observed values at each time-step with those reported by CHIRPS or the WC maps, which in the latter case meant estimating the climate variables based on the long-term averages in the WC maps."*

*Also, in Tables 4 and 6, which include the results of the sensitivity tests, we included footnotes to make this clearer.*

*Furthermore, the discussion of these results was enhanced by rewording/adapting the following paragraphs in Section 5.*

*"Furthermore, it was found that the quality of results of the GLMM are particularly sensitive to the number and location of gauges measuring temperature. As shown in Table 3, the RMSE for this approach rises sharply when only 8 (3.89 °C), 5 (3.99 °C) and 2 (14.44 °C) gauges are used to estimate its parameters. The performances of IDW and LR also decrease considerably (RMSE of 9.34 °C and 7.78 °C respectively, with only two gauges), to the extent that using the raw WC maps for this case study (RMSE of 3.36 °C) may be preferable to any method other than WCA once the density of gauges becomes low."*

*"The other precipitation interpolation approaches decrease their performance at a relatively similar rate, when facing a reduction in the number estimation gauges. As shown for the LOOCV (see Figure 7B), this may be because errors at high elevation gauges strongly influence the overall RMSE. When only 4 gauges are included, however, ChA and to a lesser extent WCA show a better RMSE (21 mm and 23.5 mm respectively), although the former has a relatively low POD (88.4 %) and the latter a larger FAR (27.9 %). It was also found that CHIRPS as a standalone product is a useful alternative to the interpolation approaches when 4 or fewer gauges are available, with only marginally worse RMSE value than IDW and better RMSE than LR and GLMM (RMSE=26.2 mm, POD=88.5 % and FAR=28.6 %)."*

**8. Include a paragraph in the methods section discussing the correlation analysis.**

*A paragraph has been included at the beginning of Section 3.1 before Table 1, briefly describing the correlation analysis and its purpose. The more in-depth discussion of the results was kept in the first paragraph of the results section (Section 4), as we consider that this is a more suitable location than Section 3. The paragraph included is as follows:*

*"Before using the covariates mentioned in Table 1 (e.g. WC, elevation, CHIRPS), an analysis of their correlation with the climate variables was done. This included plotting temperature and precipitation observations versus the covariates, and computing Pearson Correlation coefficients."*

> a. **Merge Fig 8-13 in a 6 panel figure.**

*The figures have been merged. See answer to comment 2a.*

> 9. **Section 4.1 – Explain how and why the 5 yrs daily average was calculated, and explain that this was for plotting purposes only in Figs 14-16.**

*This aggregation was done for illustration purposes only. Our goal with these figures was to show: what methods over and under-estimate observations, by approximately how much, how this changed as a function of the period of the year, and how this changed as a function of different types of stations. The 5-year series of daily data contained too much variability to visually assess the trends, which was achieved using the averaged series.*

*For the same reason, to facilitate the visualisation of the main trends, values were also smoothed using the LOESS method. Briefly, the method analyses data nearby a point X (how much data is included is a user defined parameter), and does a simple regression using this data. The value of X is adjusted to the value predicted by this regression.*

*Although this may eliminate day-to-day fluctuations, the overall trend over several days is shown much more clearly, as the noise is reduced. The LOESS is just one of the several methods that could be used to do this (a simple moving average could have also been used). A reference was provided so the reader can have access to more details (Jacoby, 2000). This information was not provided in the previous version because we did not consider it to be very relevant, taking into account that the method is only used for illustration purposes.*

*To address this, we have included the following paragraph:*

*"Figure 5 illustrates the daily temperature averaged over the 5-year period of analysis for sites 18, 27 and 28 (similar results were found for the rest of the gauges). Values were averaged in this way purely to facilitate visualisation of results, as the daily variability over the five years makes it difficult to see what approaches over and under-estimate observations, by approximately how much, and how this changes as a function of the period of the year. The performance metrics were calculated with the non-aggregated data."*

*We have also included a footnote for the figure to better explain the purpose of using the LOESS method.*

*"All curves were smoothed using the LOESS method (Jacoby, 2000) with \alpha= 0.045, this is similar to a moving average and is used to facilitate the visualisation of the main trends only"*

> a. **Merge these three in a three panel figure.**

*The figures have been merged. See answer to comment 2a.*

> 10. **Why CHIRPS data was not analysed in the same way as WC? Or be more clear how the CHIRPS data was merged with observations.**

*In the revised manuscript, CHIRPS has been used in the same way as the WC maps, in order to generate methods WCA and ChA. Table 1 and Section 3 now explain in much more detail how alternative datasets were merged with observations.*

*See also the response to comments 6 and 7B.*

*The new method ChA ended up having a very good performance, and this is mentioned in the Discussion and Conclusions.*

240

**a. Merge figures 17 and 23. 19 and 21. 18 and 22.**

*Figures 19-21, and 18 and 22 were merged. See answer to comment 2a. We consider that due to the size and content of figures 17 and 23, it is desirable to keep them separated.*
**Reviewer 2**

245 **Specific Comments**

**Introduction**

How have other authors addressed this topic? There is a strong discourse on this issue and a large number of researchers developing precipitation products as MSWEP, CHIRPS and CR2
250 have dealt with this problem. Please elaborate on the findings of other authors working with high elevation data. Also how do authors deal with missing information in hydrological modelling, which interpolation methods have worked and which were the results of evaluating different satellite based and combined precipitation data sets in data scarce Andean regions? Although you mention some authors, their findings are not described or
255 compared. Ideally, these should help to justify your objectives.

*We have made a major revision of the introduction following this comment. We have complemented the literature review with further references, and we have explained in a clearer way how other authors have used other interpolation approaches and alternative datasets within the Andean mountains, for*
260 *both temperature and precipitation. Then, we use this information to highlight the gaps in the literature, which end up supporting the scope of our paper.*

*Furthermore, we have narrowed the scope of the paper so it is clearer from the beginning what the reader can expect from the rest of the paper.*
265
*We have also explained in Section 3 in much better way, why we selected the GLMM, CHIRPS and WC data, from other approaches and alternative datasets.*

**Data**
270

The data (input, validation..) should be presented in the main text. Otherwise the numbers in the map are useless. Also in the map, it would help to enlarge it and use other colours for elevation and delineate a stronger catchment area to make the map understandable even in black and white. Numbers in the map should also be visible in Figures 2 and 3.
275
*We are not sure about what the reviewer means by including the "data (input,validation..)" in the main text. We have tried to follow general practice from similar papers working with similar data, which commonly include:*

- *A map of the region being analysed including the location of the gauges.*
280 - *A list, usually in an appendix, of the stations analysed, providing detailed information of the location, variables measured and availability of observations (this is not included when analyses involve a very large number of stations e.g. > 100).*
- *An overview of the data with some figures that shows seasonality patterns and the range of values of some of the gauges analysed..*

*Furthermore, we are not sure how we could differentiate validation stations, as a leave-one-out cross validation method was used, which means that all stations were both used for calibration and validation in different runs of the model. We have included a paragraph in Section 3.3 to better explain the LOOCV, and what this means in terms of the gauges used for validation and as input data.*

*"In order to assess the performance of the approaches, one gauge was removed from the group used to interpolate the climate variable, and the set of errors for that gauge were recorded as the difference between the interpolation results for that location and the corresponding observations. After repeating this for all gauges, the concatenated errors are used to calculate the validation metrics. This leave-one-out cross-validation (LOOCV) procedure was applied separately for temperature and precipitation and for each interpolation approach."*

*The map has been updated following the comments from both reviewers, to make sure that the catchment is easy to identify, terrain elevation is easy to differentiate and the location of the stations is clearer. Figures 2 and 3 were updated as well following comments from reviewers. Also, a CHIRPS figure was provided for the reader to visualise this product and compare it with the WC data, particularly the resolution of both within the area of analysis. We did not think it was worth including the location of gauges in the new Figure 3, as the purpose of this figure is to give an idea of the resolution of the alternative datasets thus it could be redundant to include the gauges.*

It is not well explained why you only used such a short period. There are enough data available to fill gaps (CR2 P dataset, Chirps, MSWEPv2.2, etc.). Temperature of course is difficult but at least different time periods could be compared. The main variable of interest should be precipitation. - Why do you present a spatial distribution of Chirps in May 2009 instead of comparing it with values from observed data?

*One of the added values of our project was to include high elevation data for both precipitation and temperature. The dataset we received from the private companies in the area was limited to this period thus we think it was logical to stick to it. We have tried to explain this in much more detail in the following paragraph:*

*"The period of analysis spans from September 2008 to August 2013 as the data obtained from the high elevation gauges was restricted to these years. Although not long enough to analyse long-term trends, the selected period allows testing of the interpolation approaches over both dry and wet years. Figure 2 provides an overview of the data by showing the monthly average temperature at four representative gauges over the five year period of analysis, and the monthly precipitation at three representative gauges throughout the same period (see Figure 1 for the location of these gauges)."*

*The temporal infilling using these products could have been inappropriate for creating a dataset for assessing the spatial interpolation methods. We thought that the spatial interpolation methods would be better assessed using observations from gauges only. The use of alternative spatial data sets is useful when used as inputs to the spatial interpolation methods, which he have done, as long as available observations from gauges are used to assess their proficiency. We used datasets such as the ones that the reviewer suggests, e.g. we used CHIRPS and WC maps, however, the scope of the paper was not to use all of the data sets available for the case study but to analyse the interpolation approaches. We have explained in more detail in Section 2.3 why we selected the alternative datasets we used.*

*We would like to further stress that we have not attempted to claim that the results in the paper are representative of long term trends. We have been cautious highlighting in multiple parts of the paper that our findings are restricted by the limitations of the study, however, this does not mean that they*

are not useful. We think that they provide valuable information of the performance of some methods, under a complex climatic region with few observation gauges.

**Methods section 3:**
The first paragraphs of this section should be part of the introduction as they deal with the general state of the art. - The advantage of using GLMMs and its exact output in this context is not clear to me. - There should be a conceptual figure explaining the methodology, input data and outputs - You use station data and as Covariates Chirps and ENSO as model input to test different interpolation methods. Then in the results section you correlate station data with Chirps and other data products for the station pixel? This part should be shifted to the data section and justify the method and data input (or not?). - 4.1 difference between input data and validation data not presented.

*As explained in the response to comment 4 by reviewer 1, we moved a lot of information on the state of the art of the methods and the datasets from Section 3 to the introduction. We also included a new paragraph in Section 3 better describing the correlation analysis between climate variables and covariates. However, we consider that it is better to keep the outcomes of this analysis in the results section, as they are part of the process to build the GLMM (i.e. defining the covariates to use).*

*We have also described in detail why we used the GLMM and what were the specific methodological advantages of using it in this case study. We did not include a Figure explaining the GLMM because we provided this information in Table 1, and we think it is now much more clear what are the inputs and outputs of each one of the approaches.*

*We are not sure what the reviewer means by differences between input data and validation data. By using a leave-on-out cross validation (for example as applied in Manz et al. (2016)), we believe we go a step ahead of using one part of the data for estimation purposes, and the rest for validation. We run each method several times, and in each of them we remove one station at a time, to validate the results of that specific run. We repeat this process for all stations, which means that all stations were used for estimation purposes, but at the same time each of them was used once for validation purposes. The overall output is the average results of all validation stations (i.e. all stations, but only when they were used for validation).*

*We think that the revised manuscript explains this in much more detail.*

**Results**:
In light of the above described missing information regarding the data input, validation data and output variables, it is difficult to understand the results and their interpretation. Overall presentation structure and language are still very poor. There are too many figures with little information content. Please focus on the main findings and try to present them in fewer self-explanatory figures.

*In the revised manuscript we have explained in detail the Leave-one-out cross validation, the inputs/outputs of each approach and the reasons for using the GLMM. We have also improved the structure of the manuscript taking into account the comments from reviewers, particularly the introduction, data and methods section.*

*The large number of figures was an issue in the previous version and we acknowledge this decreased the presentation quality of that version. However, as explained in the answer to the comment 2A of reviewer 1, we have solved this issue by merging lots of figures in multi-plots.*

*We have also narrowed the scope so there is consistency throughout the paper on the aims, results*
380 *and key added value of the paper.*

**References**

[revised manuscript text omitted]

---

## Referee Report (RR1)

**Comparison of approaches to interpolating climate observations in steep terrains with low-density gauging networks**

**General comments :**

The authors propose a comparative analysis of interpolation methods in a mountainous area with scarce availability of hydrometeorological information. As part of the methodology, it is proposed to compare the performance of a method of great complexity with respect to other less complex methods. As a result, the interpolation methods that can be used for hydrological modeling exercises in the specific context of this case study are defined.

In my opinion, the work can be accepted as long as an exhaustive presentation of the quality and variability of the data is made. This will allow contrasting the uncertainty of the data with the results obtained. Which in turn will allow to conclude on the reliability of interpolation methods, discuss their advantages and limitations. In order to be able to make the respective recommendations on the use of the most suitable interpolation methodology for this specific case.

For the aforementioned it is necessary to make a quantitative presentation of the degree of uncertainty in obtaining the data used. In addition, the seasonal variability of precipitation and temperature should be presented at the sites where the information is available to do so. This despite the series of temporary limitation in the high-altitude series. Additionally, it should be explained based on the literature review how regional / global climatic factors influence the variability of precipitation and temperature in the study area. Similarly, local factors that influence temperature and precipitation should be indicated, such as the location of the basins with respect to the Sun and the preferential direction of wind circulation.

This information would allow: 1) to know the degree of precision of the methods used to obtain the hydrometeorological data (Ochoa-Tocachi et al., 2018), 2) to know the magnitude and variability of the seasonal cycle of the variables analyzed in the years studied with respect to the climatology of the area (Francou, 2004), 3) to understand the factors of regional circulation (Garreaud, 2009) and the local geographic conditions that influence the variability of precipitation and temperature (Buytaert et al., 2006), 4) understand why ENSO indicators are used in the interpolation without considering a time lag due to the distance from the Pacific area, which in principle would be advisable (Francou, 2004).

This would correspond to the climate description section of the article, which has not been sufficiently extended in the current version despite having been requested by previous reviewers.

**References**

Buytaert, W., Celleri, R., Willems, P., Bièvre, B. De, Wyseure, G., 2006. Spatial and temporal rainfall variability in mountainous areas: A case study from the south Ecuadorian Andes. J. Hydrol. 329, 413–421. https://doi.org/10.1016/J.JHYDROL.2006.02.031

Francou, B., 2004. New evidence for an ENSO impact on low-latitude glaciers: Antizana 15, Andes of Ecuador, 0°28′S. J. Geophys. Res. 109, D18106. https://doi.org/10.1029/2003JD004484

Garreaud, R.D., 2009. The Andes climate and weather. Adv. Geosci. 22, 3–11. https://doi.org/10.5194/adgeo-22-3-2009

Ochoa-Tocachi, B.F., Buytaert, W., Antiporta, J., Acosta, L., Bardales, J.D., Célleri, R., Crespo, P., Fuentes, P., Gil-Ríos, J., Guallpa, M., Llerena, C., Olaya, D., Pardo, P., Rojas, G., Villacís, M., Villazón, M., Viñas, P., De Bièvre, B., 2018. Data Descriptor: High-resolution hydrometeorological data from a network of headwater catchments in the tropical Andes. Sci. Data 5. https://doi.org/10.1038/sdata.2018.80

---

## Author Response (AR2)

**Comparison of approaches to interpolating climate observations in steep terrains with low-density gauging networks**

**Response to comments by Reviewers**

Once more, we are very grateful with the two anonymous reviewers who have provided very valuable feedback to improve the manuscript. We are also pleased that only minor revisions were required, which shows the improvements of the manuscript from the previous version.

**Reviewer 1**

We are glad the reviewer found our work interesting and publishable, we address his specific comments as follows.

**Specific Comments**

1. **For the aforementioned it is necessary to make a quantitative presentation of the degree of uncertainty in obtaining the data used.**

We have updated the last paragraph of section 2.2 to provide more details about this. Many of the gauges from public institutions and one of the gauges from private companies had already been checked and used for research purposes, so we included some of those references. We also included the precision of the DGA gauges (the DGA is the government institution with the largest number of gauges in this area and in general in the country). Regarding the details of the ETH-Zurich observations of *Pellicciotti et al. (2008)*, we provided a reference to the original paper describing the data. Finally, further details of the exclusion of precipitation observations at two sites were included. The new paragraph at the end of Section 2.2 is as follows:

> *Many of the precipitation and temperature data from government institutions have previously been published (Castro et al., 2014, Jacquin and Soto-Sandoval, 2013, Vicuña et al., 2011, Ragettli et al., 2014, Zambrano-Bigiarini et al., 2016), and it has been reported that the precision of the DGA precipitation gauges is 0.1 mm (Castro et al., 2014). In addition, the characteristics of the gauges used by Pellicciotti et al. (2008) were reviewed and the gauges precision was deemed sufficient for the purposes of this project. In order to complement this, data quality was analysed using double mass plots, and plots of the relationship between elevation and the climate variables. This led to the exclusion of precipitation measurements at site 26. Furthermore, all precipitation observations at site 29 were also excluded following the suggestion from one of the mining companies, which stated that the gauge at this site had technical issues when measuring solid precipitation. The temperature measurements at both sites did not show any anomaly. No further issues with data quality were noted.*

The decision of excluding precipitation observations at site 26 was taken following an initial analysis of Elevation vs Precipitation (see Figure R1, which is similar to Figure 4F in the manuscript although it contains information for July). From this figure, it was realised that precipitation at Site 26 was very low compared to the rest, and that it did not follow the trend. Due to this inconsistency (which was repeated for all other months) and as a preventive measure, this site was excluded from further analyses.

[Figure]

**Figure R1 – Long term average July precipitation vs Elevation of gauges.**

Furthermore, although not reported in the article for brevity purposes, we checked the consistency between the most reliable gauges and the rest, particularly Site 17 (Los Bronces), which was a high elevation gauge not previously used for research. The most reliable stations (Site 7 - 05410008-6, Site 9 - 05414004-5, Site 27 - Lagunitas, Site 5 - 05410006-K, Site 2 - 05403006-1) were defined after reviewing research articles and government documents that used precipitation data and conversation with local experts that know the network of gauges. We include some of the double mass plots as follows and we also include some plots of the correlation between precipitation in multiple sites. We decided not to include these plots in the paper, consistent with the level of information provided in other papers of this type.

[Figure]

**R2 – Double mass plot between Site 10 (05414005-3) and Site 7 (05410008-6).**

[Figure]

**R3 – Double mass plot between Site 12 (05422002-2) and Site 5 (05410006-K).**

[Figure]

**R4 – Double mass plot between Site 15 (05733006-6) and Site 5 (05410006-K).**

[Figure]

**R5 – Double mass plot between Site 16 (05733010-4) and Site 5 (05410006-K).**

[Figure]

**R6 – Double mass plot between Site 17 (Los Bronces) and Site 5 (05410006-K).**

[Figure]

**R7 – Correlation of precipitation observations between Site 17 (Los Bronces) and Site 5 (05410006-K).**

[Figure]

**R8 – Correlation of precipitation observations between Site 15 (05733006-6) and Site 9 (05414004-5).**

All these analyses were done with the aim of making sure that the data used were of acceptable quality for the current research, and that there was consistency between observations. Strange double mass or correlation plots would likely have flagged issues regarding uncertainty in the observations. Apart from the issues reported for site 26, no other quality issue was found, thus we are confident that the uncertainty in the measurements is relatively low compared to the magnitude and variability of observations.

2. **In addition, the seasonal variability of precipitation and temperature should be presented at the sites where the information is available to do so. This despite the series of temporary limitation in the high-altitude series.**

The previous version of the paper already included two figures that helped understanding the seasonal variability of the temperature and precipitation (Figure 2 in the previous version of the manuscript). In Figure 2A in the previous version (Figure 2 in the new version), it was possible to see the seasonal variability of temperature for four representative gauges. Site 19 is in the valley (529 masl), site 29 is the lowest one in the mountains (1585 masl), site 27 is at 2922 masl and site 28 is the highest one at 4080 masl. We did not include all the gauges in the figure because this would have saturated the image and it would have been redundant, as the behaviour of gauges at similar elevations is not very different. Furthermore, as explained in the text, many of the gauges were only available during one summer season, which means that there was no information for the rest of the year. A figure showing the seasonal variability of all temperature gauges in the catchment is presented below, and it is possible to see that showing the values of four gauges only (as in the paper) is equally as illustrative, but much more efficient (less data conveying the same information). For the same reason, a figure of daily values over the five year period was not included either.

[Figure]

For precipitation, we have complemented the figure of monthly observations with a figure showing monthly average precipitation for three gauges, which helps understanding of the seasonal changes for this variable. In this way, the new Figure 3 contains monthly average values over the period of analysis, and also the monthly observations during the same period.

A figure showing the values of all gauges was not included for the same reasons explained for temperature. This figure is presented below, and it can be seen than once more, there is not too much added value of showing all gauges compared to the figure included in the paper, as many of them are similar, and the figure becomes saturated.

[Figure]

**3. Additionally, it should be explained based on the literature review how regional/ global climatic factors influence the variability of precipitation and temperature in the study area.**

We think we have included a decent amount of detail (i.e. similar to other papers like Castro et al. (2014) and Zambrano-Bigiarini et al. (2016)) on how global climatic factors affect the weather in the case study. At the beginning of Section 2.1 we mention the overall climate characteristics, and we mention how key global climatic factors such as the South Pacific Anticyclone and the ENSO, affect the area. This amount of detail helps understanding the climate context of the catchment, and how this is related to the observations during the period of analysis.

Regarding regional and local factors, see answer to Comment 4.

**4. Similarly, local factors that influence temperature and precipitation should be indicated, such as the location of the basins with respect to the Sun and the preferential direction of wind circulation.**

We have modified two paragraphs in Section 2 to provide more information on the local factors that may influence precipitation. We have included details of the influence on the windward position of the area and we explained the overall location of the sub-catchments, which allows understanding of their position with respect to the sun. In addition, we provided some details of the preferential direction of wind circulation, however, for the latter there is very little information at a small scale, as there are few wind observations in the area, so it is only possible to provide large-scale details.

The first paragraph at the beginning of Section 2 is now as follows:

*The Aconcagua River is an important source of water in Central Chile (Pellicciotti et al., 2007). The source is located in the Andean mountains near the border of Chile and Argentina, where most of the subcatchments run south to north (Blanco and Juncal) or north to south (Colorado). Once these streams connect and create the Aconcagua River, it flows east to west towards the Pacific Ocean. Topography fluctuates from coastal areas to peaks of approximately 5900 m above sea level. The catchment has an area of approximately 7500 km²; however, the upper*

*section, which is the subject of this research, is only around a third of this and includes the Andean mountains and a portion of the central valley (see Figure 1).*

The first paragraph at the beginning of Section 2.1 is now as follows:

*Climate within the Aconcagua catchment is Mediterranean, close to semi-arid conditions (Ohlanders et al., 2013). Annual average precipitation is approximately 350 mm, however, most of this is concentrated during the austral winter (frontal rainstorms during June, July and August), when the South Pacific Anticyclone retreats from the region (Falvey and Garreaud, 2007, Montecinos and Aceituno, 2003). This is complemented by occasional convective storms (Garreaud et al., 2009, Viale and Garreaud, 2014). Furthermore, precipitation is also highly influenced by the orographic effects, with the windward slope of the Andes having much more precipitation during winter, compared to the leeward side (Viale and Garreaud, 2015, Viale and Nuñez, 2011). There are few wind gauges in the area, which hinders understanding of the small-scale preferential wind directions and their influence on precipitation. However, it has been reported that at a macro scale the prevalent direction is west to east (Castro et al., 2014, Garreaud et al., 2009).*

5. **Understand why ENSO indicators are used in the interpolation without considering a time lag due to the distance from the Pacific area, which in principle would be advisable (Francou, 2004). This would correspond to the climate description section of the article, which has not been sufficiently extended in the current version despite having been requested by previous reviewers.**

Following the suggestion from Reviewer 1, we checked the work done by Francou et al (2004) and tested the correlation between precipitation and the ENSO index with different lags. In the aforementioned publication the authors used a 3 months lagged ENSO index, to analyse the impacts of ENSO on low-latitude glaciers. Here, we analysed the correlation between the 1, 2, 3 and 4 months lagged ENSO index and the climate variables. However, we did not find any relevant difference with the values already reported in the paper. The following sentence in the second paragraph of Section 4.1 was modified.

*The $\rho$ for monthly precipitation and WC values is lower but significant ($\rho = 0.62$), while monthly correlation with elevation is above 0.6 for most months. ENSO shows a weak correlation with precipitation $\rho = 0.12$ (not shown in Figure 4 for brevity purposes), and similar results are obtained with time-delayed (1 - 4 months) ENSO index (i.e. as done in (Francou et al., 2004)).*

It is important to mention that at the end of the second paragraph on Section 4.1, we provided a brief explanation of why this correlation may not have been very strong in this project.

**Reviewer 2**

We are grateful with Reviewer 2 for highlighting the value of the paper and for their suggestions. A detailed review of his or her specific comments is provided below.

**Specific Comments**

**Page - Line – Comment**

- **2 – 29 – "KED" has not been defined above. Please explain what this acronym means.**
Acronym included in the third paragraph of the introduction.

- **3 – 1 – The part in parenthesis … "(except in dry areas… )" is repeated below in the same sentence. Remove?**
Despite this may sound redundant, we wanted to be clear that in dry areas, satellite products struggle to monitor precipitation detection (i.e. occurrence) and their magnitudes, these are two different, although related, things. As opposed to this, for extremely wet areas, satellites tend to have problems with the magnitudes, but not with the detection.

- **3 – 6 to 11 – The Acronyms PGFv3, CHIRPS, TMPA, etc have not been defined. Please provide appropriate references.**
A reference to each one of the datasets has been included, and a full explanation of CHIRPS, which is the only dataset used in this research, is provided later in the paper.

- **3 – 19 - … precipitation MAGNITUDES and temperature…**
Corrected

- **4 – 5 – The "source" is located… Change to: The "western limit of the basin" is located in the Andean mountains "along" de border…**
We included the word "*along*", however, we think that it is better to say "the source", as readers that do not know the local context may not know that the western limit is the source of the catchment.

- **4 – 13 – The Anticyclone "retreats"… Explain better?**
We used the term "retreat" based on the jargon used in (Falvey and Garreaud, 2007), to explain that the Anticyclone moves away from central Chile during winter, allowing frontal rainstorms to get to the area and generate precipitation.

- **4 – 19 – There is considerable inter-annual variability… of what variables?**
Although there may be more references analysing the inter-annual variability of precipitation, there can be changes in temperature as well, thus we decided to update the sentence as follows to make this clear:
    *"There is relevant inter-annual variability in the climate variables related to El nino …."*

- **5 – Figure 1: Put Argentina and Chile in the second inset. Put also "Aconcagua river BASIN" and "Maipo river BASIN" in the map (the rivers are not shown, just their upper basins).**

We included the word Catchment for both Aconcagua and Maipo. We do not think we should include more text in the smaller maps, as they are quite small and more text may make them look very crowded.

- **6 – 11 – "long-term gauges" which ones? List them in the table**

We have edited this section and made sure that the legend in the figure with the name of the gauges is clear enough so readers can check the number of the site and the name of the gauge, with Figure 1 and the table in the appendix.

- **7 – Figure 2. These diagrams show different things for different variables (annual cycle for temperatures, and monthly time series for precipitation). This is confusing and leaves the reader with the feeling that something is missing. We cannot see the temperature changes over the 2008-2013 study period, nor the annual cycle of precipitation in the basin. I believe it would be better to show the annual cycle for both T and PP, and below the monthly time series for both T and PP. Please show this to characterize the general climate variations and seasonality in the basin.**

We have separated the figures as suggested by the reviewer and Figure 2 now shows the monthly average temperature, while Figure 3 shows monthly average precipitation and monthly precipitation observations. Due to the number of observations (daily values) for temperature, and the fact that this variable change less from year to year, compared to the year to year changes of precipitation, we deemed there was not too much added value of including this figure for temperature.

- **8 – Figure 3. This figure is not clear either. On the left we see the actual monthly CHIRPS data for May, and on the left the May climatology from Worldclim. At a first look these maps are quite different, not only due to the pixel resolution but also because their values for different sectors of the basin not consistent at all. This makes the reader believe the CHIRPS data is of little use, when in reality this is not the case. Why not showing the May climatology (long-term average) for CHIRPS too? Or at least make the difference between the maps more clear in the legend and in the maps themselves.**

This is the second time that we receive comments about this figure and it seems that we did not manage to convey the message we wanted (i.e. the comparison of the resolutions between WorldClim and Chirps data). In addition, as stated by this examiner, we may be confusing the reader. Due to this, we thought it would be better to remove this figure.

- **11 – 17 - …for this VARIABLE…**

Corrected

- **14 – 3 to 6 – Please discuss and assess the various diagrams in Figure 4 in order (first fig 4A, then Fig 4B, and so on). You should also mention that the diagrams with the low correlations between ENSO and T are not shown in Fig 4.**

We think the discussion in Section 4.1 is concise but illustrates the key points that affect further analyses in the paper (e.g. Excluding ENSO for the temperature GLMM). We also provide a short analysis on why the correlation with ENSO is not stronger. Other findings, like the improvement on the CHIRPS correlation at a monthly scale, follow our findings in the literature review, which are stated in our introduction (see the last lines of paragraph 5 of the introduction).

We tried keeping this part as concise as possible to focus on the key results of the paper, thus we think we should not add any more content here.

- **14 – 9 to 11 – Here you should also mention that the diagrams with the correlations between ENSO and PP are not shown in Fig 4.**

We have modified a sentence in the second paragraph of Section 4.1 to address this

*"ENSO shows a weak correlation with precipitation ρ= 0.12 (not shown in 4 for brevity purposes), however, a monthly analysis shows that for several months the correlation is close to ρ = 0.5, therefore, it was decided to keep ENSO as a covariate for the precipitation GLMM."*

- **14 – 16 – (23 TEMPERATURE gauges)**

We realised this was a bit redundant and it had been already explained, and it could confuse the reader as the there were 24 temperature gauges in total, although the LOOCV was done in groups of 23. Due to this, the "(23 gauges)" was eliminated.

- **19 – Table 6. In the note below the table, Shouldn´t it should say … provided by WC to approximate daily PRECIPITATION at all sites…**

Corrected

**References**

[revised manuscript text omitted]

---

## Author Response (AR3)

**Comparison of approaches to interpolating climate observations in steep terrains with low-density gauging networks**

**Response to comments by the Editor**

**Page 7 line 8: Footnotes should be avoided according to the guidelines (https://www.hydrology-and-earth-system-sciences.net/for_authors/manuscript_preparation.html), and in addition, the html link will not work in the print version. Please put in text body and use reference style for Webpages (see "References" in the guidelines).**

Done

**Page 14 line 28: Citation should be "Francou et al., 2004", please check with the guidelines for in-text citations.**

We have updated the Copernicus Latex package with the most recent version following instructions from Viola Zierenberg, but after re-running the code we obtain the same type of in-text reference (i.e. \citet{} in Latex). We further contacted Viola Zierenberg about this issue and we obtained the following reply.

*Dear Juan,*

*Thank you for your email and for further explaining the issue.*

*Citations that are part of the sentence (as in your example) should be inserted with the "citet" command (**Note from us: This is the command we have been using so far**):*

*…as done in Francou et al. (2004)…*

*…as done in \citet{Francou2004}…*

*Citations that are not directly part of the sentence should be inserted with the "citep" command*

*…as done in another publication (Francou et al., 2004).*

*…as done in another publication \citep{Francou2004}. ("[e.g.][]" can be used here as well)*

*…as done in another publication (\citealp{Francou2004}).*

*Please note that citations are often corrected during the typesetting process and adjusted to our house standard. Therefore, I believe it is acceptable if you adjust the citations to our standard even if this might contradict the editors comment.*

*If you have any further questions, please do not hesitate to contact me.*

*Kind regards,*

*Viola*

Would it be possible to accept the references as they are in the text and should there was any correction required, the typesetting process may be able to help?

**Page 25, competing interests: Please use the wording suggested in the guidelines ("The authors declare that they have no conflict of interest.")**

Done

**Figure 3B and Figure 8: For clarity please place major grid lines and labels at the beginning of the year (southern hemisphere) instead of Nov/May and May, respectively (unless there is a compelling reason to place them where they are currently.)**

Done. Labels have been inserted every January during the period of analysis, while vertical break lines were included every 6 months.